# COMMUNICATION-EFFICIENT FEDERATED LEARNING VIA MODEL-AGNOSTIC PROJECTION ADAPTATION

## ABSTRACT

Federated learning (FL) enables collaborative model training across distributed clients without centralizing sensitive raw data while benefiting from diverse data sources. Despite recent advancements in FL, the communication overhead remains a significant challenge, especially for large-scale models. Recent low-rank adaptation (LoRA) techniques have shown promise in reducing these burdens in FL, but they are typically applied to each layer individually and depend on the model architecture, which limits their performance. To address these shortcomings, we propose Model-Agnostic Projection Adaptation (MAPA), a novel approach that applies factorization to the entire model parameter space, which we view as a *single vector*, regardless of the number of layers and model architecture. MAPA factorizes the single-vector model update into a fixed *reconstruction matrix* and a trainable *projection vector*, with the reconstruction matrix being randomly initialized using a shared seed at each round. This ensures that *only* the projection vectors need to be communicated to the server, thereby reducing the communication cost. Furthermore, MAPA's vector-based representation and relaxed rank constraints allow for a larger reconstruction matrix and smaller projection vector dimensions compared to LoRA, enhancing the expressiveness of model updates while significantly reducing communication overhead. Experimental results demonstrate that MAPA outperforms existing FL methods in both communication efficiency and model performance, effectively coupling optimization and communication efficiency in FL environments.

## 1 INTRODUCTION

Federated learning (FL) is a distributed machine learning framework that enables model training across numerous devices, referred to as clients, without the need to collect or process client data on a server. In a typical FL process, each client downloads an initialized model from the server, trains it using local data, and then uploads the updated model back to the server. The server aggregates these updates to refine the global model, employing techniques such as federated averaging (FedAvg) (McMahan et al., 2017). This iterative process is repeated over multiple communication rounds, enabling clients to improve the model collaboratively without data sharing.

Despite the notable benefits of FL, a primary challenge is the substantial communication overhead involved in transmitting model updates between clients and the server, especially when dealing with resource-constrained clients and large-scale models with numerous parameters. This communication overhead can become a significant bottleneck, limiting the scalability and efficiency of FL.

To address the communication burden in FL, various strategies have been developed that focus on reducing either the communication frequency or the communication load per round. To decrease communication frequency, methods such as performing multiple local epochs on clients (Stich, 2018) and selecting a subset of clients to participate in each training round (Sattler et al., 2019; Li et al., 2020) have been proposed. On the other hand, methods aiming to reduce the communication load per round have been more extensively studied. Konečný (2016) broadly classified these methods into two categories: (i) *sketched updates*, where the local model is first optimized, and then the update is compressed before transmission, and (ii) *structured updates*, where the model is optimized in a subspace with fewer trainable parameters, which are then transmitted to reduce communication. These strategies are complementary and can collectively contribute to enhancing the scalability and efficiency of FL.

Figure 1: Overview of the MAPA method. Unlike existing LoRA approaches, MAPA treats the entire model parameters as a single vector before factorization. This allows MAPA to use a larger reconstruction matrix $A$ and a smaller dimension for the projection vector $B$, leading to more efficient FL. MAPAX further generalizes this idea by trading off communication, computation, and memory through partitioning and parallelization.

Low-rank adaptation (LoRA) (Hu et al., 2021; Ou et al., 2023; Bertsimas et al., 2023) is a popular structured update method that decomposes parameter updates for each layer independently as $\Delta w_{d_1 \times d_2} \approx A_{d_1 \times q} B_{q \times d_2}$, with the rank constrained by $q \leq \min(d_1, d_2)$. Recently, many researchers have applied LoRA in FL to enhance training efficiency Yi et al. (2023); Sun et al. (2024); Cho et al. (2024); Kuo et al. (2024); Yang et al. (2024); Qi et al. (2024). However, the layer-wise approach and the rank constraint in LoRA restrict the ability to fully capture the low-rank structure of the global gradient, thereby limiting the performance of these methods.

This paper proposes **Model-Agnostic Projection Adaptation (MAPA)**. MAPA treats the entire model parameters as a single vector and factorizes the model update $\Delta W_{d \times 1}$ into a fixed *reconstruction matrix* $A_{d \times p}$ and a trainable *projection vector* $B_{p \times 1}$, where $d$ denotes the number of model parameters and $p \leq d$ is the reduced dimension. In contrast to *Freeze A LoRA* (FA-LoRA) methods Sun et al. (2024); Zhang et al. (2023); Zhu et al. (2024); Hao et al. (2024), the reconstruction matrix is initialized randomly with a shared seed on every FL round, and it is not frozen during training. Our approach still eliminates the need to transmit $A$ and limits the communication to the projection vectors. Compared to LoRA-based methods, MAPA's vector-based representation and relaxed rank constraints allow for a larger reconstruction matrix $A$ and smaller projection vector $B$ dimensions, enhancing the expressiveness of model updates while reducing communication costs.

The high compression rate of MAPA comes from its large expressive capacity by relaxing the low-rank condition $q \leq \min(d_1, d_2)$ of LoRA and factorizing the gradient signal into a single vector. However, this incurs the overhead of generating a larger reconstruction matrix $A$, which results in a higher memory and computation burden on clients. Motivated by this, we also propose an extension to MAPA called MAPAX, which mitigates this overhead and balances the trade-offs between communication, computation, and memory costs depending on the client's resources. Additionally, we show that MAPAX can cover the whole space of communication-efficient factorization, bridging the gap between various techniques and fostering a better understanding of their methods. Figure 1 visualizes the architectural differences between these methodologies in matrix manipulation forms.

Overall, we make the following key contributions:

- **Introduction of MAPA.** We present MAPA, a novel matrix factorization that operates independently of the model architecture. By treating the entire model parameter as a vector, MAPA constructs a larger reconstruction matrix, resulting in an expressive subspace that requires fewer trainable parameters than low-rank layer-wise methods.

- **Enhancement of Communication Efficiency in FL.** By integrating MAPA into FL, we achieve substantial reductions in communication by optimizing in a lower-dimensional subspace.

- **Extension to MAPAX.** We introduce MAPAX, an extension of MAPA, to address the computational and memory overhead associated with the larger reconstruction matrix. MAPAX creates a trade-off between communication, computation, and memory costs, making it adaptable to clients with varying resource constraints. We show that MAPAX bridges the gap between different factorization techniques, offering a unified understanding and approach.

- **Theoretical Analysis.** We provide a thorough theoretical analysis establishing the convergence of MAPA. We also show that MAPA outperforms LoRA-based methods in maintaining training performance while reducing communication costs.

- **Empirical Evaluation.** We conduct extensive experiments on diverse datasets and model architectures, showing that MAPA surpasses SOTA methods in both communication efficiency and model performance.

## 2 RELATED WORKS AND BACKGROUND

Among techniques introduced to alleviate the communication overhead in FL, in this section, we first explore the sketched update methods that project the gradient signal into a subspace, highlighting the similarity of these techniques to matrix factorization so we can argue further why a structured update can exploit a better gradient signal with this formulation. Afterward, we look into structured update techniques and focus on low-rank adaptation methods studied in communication-efficient FL to highlight the novelty and advantages of our work compared to recent studies.

**Sketched Update** is a two-step method, where first, the full space gradient is computed, and second, it is projected into a subspace. It includes techniques such as sparsification (Konečnỳ, 2016), quantization (Alistarh et al., 2017; Mao et al., 2022), and gradient subspace projection (Azam et al., 2021; Oh et al., 2022; Park & Choi, 2023), random subspace projection (Rahimi et al., 2024; Shi & Eryilmaz, 2021). The concept of subspace projection methods is that for a given gradient $\mathbf{g} \in \mathbb{R}^d$, reconstruction matrix $A \in \mathbb{R}^{d \times p}$, find a projection vector $B \in \mathbb{R}^p$, which minimize the compression error $\|\mathbf{g} - AB\|_2$, where $d$ denotes the total number of model parameters and $p \ll d$ is the size of projection vector.

$$B^* = \arg \min_{B \in \mathbb{R}^k} \|\mathbf{g} - AB\|_2 \quad ; \quad B^* = (A^\top A)^{-1} A^\top \mathbf{g}$$

However, solving this exact linear system can be computationally expensive, especially when $k$ is large as the exact solution has $\mathcal{O}(k^2 n + k^3)$ time complexity and $\mathcal{O}(k^2)$ memory complexity. Therefore, most works in the literature opt for approximation methods instead of solving the exact problem due to these computational challenges:

$$B^* \approx \mathbf{A}^\top \mathbf{g}$$

Given this formulation, we notice that low-rank factorization solves a similar problem. However, unlike subspace projection methods, the projection vector $B$ is computed independently of the gradient $\mathbf{g}$ by training from the data:

$$B^* = B + \eta \nabla_B \mathcal{L} \left( W + AB; \mathcal{D}_i \right).$$

Although sketched methods benefit from accessing a high-quality gradient $\mathbf{g}$, one of their shortcomings is blindness to the loss surface $\mathcal{L}(W; \mathcal{D})$ and alternative solutions beside $\mathbf{g}$ that might be more suitable for projection in their subspace. They typically perform well given a large enough $p$. However, as the compression rate increases, the reconstruction of the projection vector ends up far enough from the gradient $g$, leading to no convergence. In contrast, direct subspace optimization leverages the complete data information to find the possible solutions within the subspace, ultimately leading to a more effective reduction in loss, even with significantly smaller $p$. Figure 2 shows a simple example of MNIST training on a single node, which highlights the performance drop of sketched update techniques such as EvoFed (Rahimi et al., 2024) and Top-k Sparsification (Konečnỳ, 2016) compared to structured update such as FA-LoRA Sun et al. (2024); Zhang et al. (2023); Zhu et al. (2024); Hao et al. (2024) and MAPA, as the sparsity level increases. LoRA and MAPA can still converge, having 2 or 4 trainable parameters from space with 11274 dimensions, which is insufficient for EvoFed and Top-k to converge.

**Structured Update** is a single-step method where instead of computing the full space gradient, it restricts parameter space, reducing the number of trainable parameters needed to be calculated and communicated, including low-rank adaptation (LoRA) (Cho et al., 2024; Sun et al., 2024; Kuo et al., 2024; Yi et al., 2023; Yang et al., 2024; Qi et al., 2024), pruning (Luo et al., 2017; Zhang et al., 2018), and weight-sharing (Ullrich et al., 2017).

The LoRA is a form of low-rank approximation (Liu et al., 2022; Wang et al., 2018; Jaderberg et al., 2014; Lebedev et al., 2014; Denil et al., 2013), which is widely used because of its solid theoretical foundation and ease of hardware implementation. The common practice for a low-rank

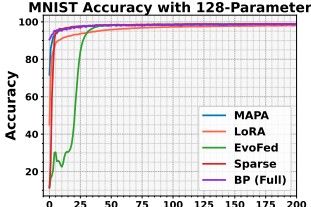
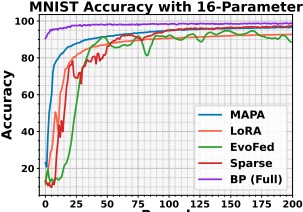
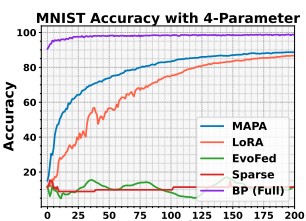
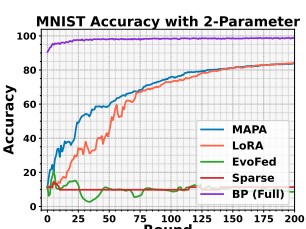

Figure 2: Performance comparison for various trainable parameters on MNIST dataset.

approximation is to approximate each layer's large-weight tensors by the product of smaller ones, reducing the rank and, consequently, the number of trainable parameters of each layer. Therefore, this factorization is dependent on the layer's architecture and requires a careful network design that considers a specific factorization for each layer.

In contrast, our technique will introduce a novel black-box factorization independent of the model architecture, not only simplifying the implementation but also performing better as it consists of a higher representation at the same rate of communication. This factorization reshapes the entire parameter matrix to the form of a single vector as $W_t \in \mathbb{R}^{d \times 1}$. Therefore, the update matrix $\Delta W \in \mathbb{R}^{d \times 1}$ is computed as $\Delta W = AB$, where $A \in \mathbb{R}^{d \times p}$ and $B \in \mathbb{R}^{p \times 1}$ having $p \ll d$.

In contrast to *Freeze A LoRA* methods, (Sun et al., 2024; Zhang et al., 2023; Zhu et al., 2024; Hao et al., 2024), we initialized $B = \mathbf{0}$ and $A \sim \mathcal{N}(0, I)$, at the beginning of each round, we update model parameters $W$ and reset $B$ and generate $A$ afresh and independently. This allows the exploration of various subspace configurations without any communication overhead and performance improvement (See Appendix B). The sub-optimality of having a frozen $A$ was also discussed in Guo et al. (2024), although we provide an alternative solution from Guo et al. (2024), which does not require training and transmission of matrix $A$, thus preserving communication efficiency. The next difference between MAPA and LoRA-based methods lies in the condition of the factorization rank. For a given matrix $W \in \mathbb{R}^{d_1 \times d_2}$, LoRA aims to reduce the number of parameters by factorizing the update as $\Delta W = AB$, where $A \in \mathbb{R}^{d_1 \times q}$ and $B \in \mathbb{R}^{q \times d_2}$, requiring the factorization rank to satisfy $q < \min(d_1, d_2)$. However, since the size of the random matrix $A$ does not add communication overhead, we focus on keeping the projection vector $B$ smaller than model parameters $W$.

To summarize, we introduce a unique matrix factorization method to reduce communication overhead in FL. In contrast to compression techniques, our approach optimizes a low-dimensional projection vector directly in the subspace, demonstrating greater effectiveness than projecting already computed gradients, especially in low-bandwidth scenarios. In comparison to existing low-rank factorization techniques, MAPA enables a much larger reconstruction matrix by treating model parameters as a single vector, relaxing the low-rank condition, and employing a model-agnostic factorization independent of the number of layers and their architecture. Finally, MAPA enhances the subspace exploration by initializing the reconstruction matrix at each turn. All contributions collaboratively result in a more expressive subspace where less information needs to be communicated, achieving greater flexibility, performance, and communication efficiency in FL.

## 3 PROPOSED METHOD

In this section, we present MAPA, MAPAX, and their application in FL. We begin by elaborating on the MAPA factorization technique, demonstrating the theoretical basis for proving its higher representation capacity while facilitating lower gradient dimensions. Then, we explain how MAPAX can be seen as the general factorization form and discuss its benefits. Subsequently, we describe the detailed process for effectively leveraging MAPA factorization within the FL framework.

### 3.1 MODEL-AGNOSTIC LOW-RANK ADAPTATION (MAPA)

Recent literature studied the effect of low-rank factorization on FL communication efficiency (Sun et al., 2024; Zhang et al., 2023; Zhu et al., 2024; Hao et al., 2024). In each layer $W \in \mathbb{R}^{d_1 \times d_2}$, the idea of LoRA is to factorize the model update as $\Delta W = AB$, where $A \in \mathbb{R}^{d_1 \times q}$ and $B \in \mathbb{R}^{q \times d_2}$ for $q < \min(d_1, d_2)$. They take advantage of freezing the reconstruction matrix $A$, limiting the trainable parameters to projection matrix $B$, thus reducing communication. While low-rank factorization shows a promising direction in FL, MAPA aims to answer a more general question: *How can we design a factorization that achieves higher representation capacity with lower trainable parameters?*

**MAPA Intuition and Description.** MAPA works toward a factorization resulting in a large reconstruction matrix and small projection matrix, leveraging the fact that random reconstruction matrices do not need to be communicated, achieving higher representation capacity without communication overhead, and resulting in a smaller projection matrix needed to be communicated. An analogy for this purpose can be seeing the reconstruction matrix $A$ as a shared vocabulary and the size of the projection matrix $B$ as the number of words used to communicate a message. A richer vocabulary (larger $A$) allows for conveying complex ideas more concisely, reducing the number of words (smaller $B$) needed to be communicated. To achieve this, MAPA treats the entire update of the model as a *single vector* and applies a black-box factorization, regardless of the number of layers or the network architecture. Let $d$ denote the total number of parameters across all layers of the

model. As illustrated in Figure 1, MAPA decomposes the *universal vector* $\Delta W \in \mathbb{R}^{d \times 1}$ into a reconstruction matrix $A \in \mathbb{R}^{d \times p}$ and a projection vector $B \in \mathbb{R}^{p \times 1}$, where $p \leq d$.

**MAPA Properties.** We aim to show that MAPA constructs a more expressive subspace, allowing a smaller $B$ to convey sufficient information for updating the model. We begin by formally defining the *communication overhead rate* ($\mathrm{CO}_{\mathrm{rate}}$) and *representation capacity rate* ($\mathrm{RC}_{\mathrm{rate}}$), in context of matrix factorization in Definition 1 and 2. Based on the established definitions, Proposition 1 and 2 formulate shortcomings of traditional factorization, and as a result, we can conclude the properties of superior factorization in the context of communication-efficiency, which finally leads to the proof of MAPA factorization superiority as shown in Theorem 1.

**Assumption 1 (Full Rank Property of Gaussian Random Matrices).** *Let $A \in \mathbb{R}^{m \times n}$ be a random matrix with entries drawn independently from a Gaussian distribution $\mathcal{N}(0, \sigma^2)$. Then, $A$ is almost surely of full rank, i.e., $rank(A) = \min(m, n)$, as the probability of $A$ being rank deficient is zero. This result follows from standard properties of random matrices Vershynin (2018); Tao (2012).*

**Definition 1 (Communication Overhead).** *Let $\Delta W \in \mathbb{R}^{d_1 \times d_2}$ be the update matrix of a model. Suppose a factorization operator $\mathcal{F}(.)$ decomposes $\Delta W$ as $\Delta W = AB$, where $A \in \mathbb{R}^{d_1 \times q}$ is a fixed random matrix and $B \in \mathbb{R}^{q \times d_2}$ is a trainable matrix. The **communication overhead** is defined as the ratio of the size of $B$ to the size of $\Delta W$:*

$$\mathrm{CO}(\Delta W, \mathcal{F}) = \frac{\mathrm{size}(B)}{\mathrm{size}(\Delta W)} = \frac{q}{d_1}.$$

**Definition 2 (Representation Certainty).** *Using the same factorization as in Definition 1. The **representation certainty** is defined as the inverse of the error rate variance. The error rate measures the expected error of the factorization to represent the original matrix, given a full-rank matrix $A$ (Assumption 1). The error expectation and variance are defined as:*

$$\mathbb{E}_A \left[ \|W - AB\|_2^2 \right] = \left( 1 - \frac{q}{d_1} \right), \mathbf{Var}_A \left[ \|W - AB\|_2^2 \right] = \left( \frac{2q(d_1 - q)}{d_1^2(d_1 + 2)} \right)$$

*Therefore, given a constant communication overhead and error expectation $\frac{d_1}{q} = r$ we have:*

$$\mathrm{RC}(\Delta W, \mathcal{F}) = \frac{1}{\mathbf{Var}_A[\mathbf{E}]} = \frac{r^3 q + r^2}{2(r - 1)} \propto q$$

**Proposition 1 (Relaxed Low-Rank Factorization Superiority).** *Let $\Delta W \in \mathbb{R}^{d_1 \times d_2}$ be the update matrix of one layer, factorized in low-rank as $\Delta W = AB$, where $A \in \mathbb{R}^{d_1 \times q}$ is a shared random matrix and $B \in \mathbb{R}^{q \times d_2}$ is the trainable matrix, with $q \leq \min(d_1, d_2)$ being the factorization rank, By reshaping $\Delta W$ into $\Delta W' \in \mathbb{R}^{(d_1 d_2)/k \times k}$ for some integer $k < d_2$, the factorization of $\Delta W'$ can achieve a higher **representation certainty** while requiring same **communication overhead** compared to the conventional low-rank factorization of $\Delta W$.*

**Collorary 1 (Single-Vector Factorization Superiority).** *Using the same factorization as in Proposition 1 for $k = 1$. $\Delta W$ reshapes into a single-vector form $\Delta W' \in \mathbb{R}^{d_1 d_2 \times 1}$ and factorizing $\Delta W'$ can achieve a higher **representation certainty** while requiring the same **communication overhead** than the conventional low-rank factorization of $\Delta W$.*

**Proposition 2 (Layer-Independent Factorization Superiority).** *Let $\Delta W_i \in \mathbb{R}^{d_1^i \times d_2^i}$ be the update matrix of the $i$-th layer of a model, and let $\Delta W_i' \in \mathbb{R}^{d_1^i d_2^i \times 1}$ be its reshaped single-vector form. In single-vector factorization methods, $\Delta W_i'$ is factorized as $\Delta W_i' = A_i B_i$, where $A_i \in \mathbb{R}^{d_1^i d_2^i \times q_i}$ and $B_i \in \mathbb{R}^{q_i \times 1}$, with $q_i \leq d_1^i d_2^i$. By concatenating the reshaped weights $\Delta W_i'$ into $\Delta W' \in \mathbb{R}^{d \times 1}$, where $d = \sum_{i=1}^n d_1^i d_2^i$. The factorization of $\Delta W'$ can achieve a higher **representation certainty** while requiring the same **communication overhead** than the conventional single-vector factorization methods applied separately to each layer.*

**Theorem 1 (MAPA Factorization Superiority).** *Let $\Delta W_i \in \mathbb{R}^{d_1^i \times d_2^i}$ be the update matrix of the $i$-th layer of a model, and let $\Delta W = \mathrm{vec}(\Delta W_1, \Delta W_2, \ldots, \Delta W_n) \in \mathbb{R}^d$ be the concatenation of all $\Delta W_i$, where $d = \sum_{i=1}^n d_1^i d_2^i$. MAPA factorization can achieve a higher **representation certainty** while requiring the same **communication overhead** than other factorizations of $\Delta W$.*

*Proof.* Collorary 1 is the result of Proposition 1 for $k = 1$. The proofs for Definitions and Propositions are given in Appendix C. Now, given MAPA is a layer-independent single-vector factorization, the proof of Theorem 1 can directly be concluded from Proposition 2 and Collorary 1. $\qquad\square$

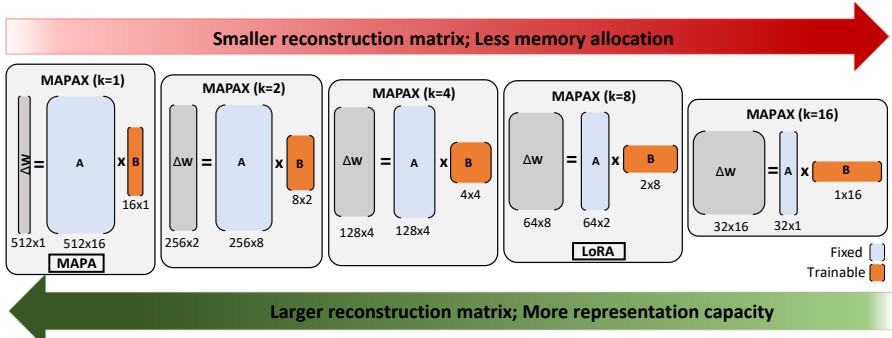

Figure 3: Illustration of MAPAX$_k$. MAPAX$_k$ reduces to MAPA when the number of partitions is $k = 1$. LoRA also becomes a special case of MAPAX$_k$ when the model is partitioned according to the layer sizes.

Thus, MAPA provides a superior representation capacity for the same communication cost. This advantage becomes increasingly significant in models with more layers or when there is a more considerable disparity between the dimensions $d_1$ and $d_2$, particularly beneficial for large-scale models.

### 3.2 MAPAX$_k$: EXTENSION WITH $k$-PARTITIONING

Building upon Proposition 1 and 2, we extend MAPA to a general form, termed MAPAX$_k$. Figure 3 illustrates the MAPAX$_k$ concept. Consider the update matrix at step $t$ as $\Delta W_t \in \mathbb{R}^{e \times k}$, where $e = \left\lceil \frac{d}{k} \right\rceil$ and $d$ is the total number of parameters representing the model's current state. To accommodate the dimensions, $\Delta W_t$ includes zero padding of size $ek - d$. The MAPAX$_k$ factorization of $\Delta W$ is then given by $\Delta W = AB$, where $A \in \mathbb{R}^{e \times p}$ and $B \in \mathbb{R}^{p \times k}$, with $p \leq e$.

Thus, MAPA can be considered a special case of MAPAX$_1$, maximizes the shared information in the reconstruction matrix $A$, and minimizes the size of the error variance. In contrast, FedAvg is MAPAX$_d$ and the rank of the reconstruction matrix $p = 1$, resulting in the projection matrix $B \in \mathbb{R}^{1 \times d}$ presenting the entire model update.

Proposition 3 states MAPAX$_k$ covers all degrees of factorization, including low-rank, resulting in a flexible approach for balancing memory allocation and representation certainty. Collorary 2 shows the case of a single layer model $W \in \mathbb{R}^{d_1 \times d_2}$, where MAPAX$_{d_2}$ is equivalent of low-rank factorization. Collorary 3 shows the case of a model having $n$ identical shaped layers, $W \in \mathbb{R}^{d \times d}$, where MAPAX$_{nd}$ is equivalent low-rank layer-wise factorization.

**Proposition 3 (MAPAX Generalization).** *Let $\Delta W_i \in \mathbb{R}^{d_1^i \times d_2^i}$ be the update matrix of the $i$-th layer of a model, and let $\Delta W = \text{vec}(\Delta W_1, \Delta W_2, \ldots, \Delta W_n) \in \mathbb{R}^d$ be the vectorization (concatenation) of all $\Delta W_i$, where $d = \sum_{i=1}^n d_1^i d_2^i$. MAPAX$_k$ factorization allocates $k^2$ times less memory for the same **communication overhead** and **error rate**, for the cost of $k$ times worse **representation certainty**, in other words, more $k$ times more error rate variance.*

**Collorary 2 (MAPAX-LoRA Special Case Single Layer).** *Let $\Delta W \in \mathbb{R}^{d_1 \times d_2}$ be the update matrix of the single layer model factorized in LoRA methods as $\Delta W = AB$, where $A \in \mathbb{R}^{d_1 \times q}$ and $B \in \mathbb{R}^{q \times d_2}$, with $q \leq \min(d_1, d_2)$ is equivalent to the MAPAX$_{d_2}$.*

**Collorary 3 (MAPAX-LoRA Special Same Layers).** *Let $\Delta W_i \in \mathbb{R}^{d \times d}$ be the update matrix of the $i$-th layer of a model with $n$ layers factorized in LoRA methods as $\Delta W_i = A_i B_i$, where $A_i \in \mathbb{R}^{d \times q}$ and $B_i \in \mathbb{R}^{q \times d}$, is equivalent to the MAPAX$_{nd_2}$.*

The proof of Proposition 3 located in Appendix C. Figure 3 illustrates this equivalency of Collorary 2, while we can conclude Collorary 3 from Figure 1.

Therefore, this extension facilitates further studies to understand better how different factorizations impact performance and total communication cost. It serves as a bridge between layer-wise or partitioned factorizations and complete model-agnostic factorizations. Furthermore, Appendix E shows complexity analysis and how to balance memory, communication, and performance.

### 3.3 APPLICATION TO COMMUNICATION-EFFICIENT FEDERATED LEARNING

This subsection explains how the factorization outlined in Section 3.1 is utilized in FL, dividing the procedure for clarity. Figure 4 visualizes the outline of this procedure.

Figure 4: Application of MAPA to communication-efficient FL.

**Matrix Construction and Broadcasting.** To ensure consistency across the network, the server and all clients start from an identical condition at each round. We guarantee identical model parameters $W_t$ and reconstruction matrix $A_t$ by broadcasting a random seed $r_t$ and the aggregated projection vector $\bar{B}_t$ at the beginning of round $t$. The initial aggregated projection vector is set to $\bar{B}_0 = \mathbf{0}$.

**In the first round** ($t = 0$), all clients and the server initialize the model $W_0$ randomly. The reconstruction matrix $A_0 \in \mathbb{R}^{d \times p}$ is generated with random Gaussian entries, and the local projection vector $B_0^i \in \mathbb{R}^p$ is set to zero, where $i$ indicated the $i$-th client and $d$ denotes the total number of model parameters and $p \ll d$ is the chosen reduced dimension.

**In subsequent rounds** ($t \geq 1$), clients update their local model $W_t$ using the previous round's matrix $A_{t-1}$, the model parameters $W_{t-1}$, and the broadcasted projection vector $\bar{B}_t$ as follows:

$$W_t = W_{t-1} + A_{t-1}\bar{B}_t. \tag{1}$$

Afterwards, clients generate a new $A_t$ by sampling from a Gaussian distribution $\mathcal{N}(0, I_{d \times p})$ using the random seed $r_t$ and set $B_t^i \leftarrow \mathbf{0}$. This ensures that $A_t$ and $W_t$ are synchronized and updated.

**Local Optimization of Projection Vector.**

---

**Algorithm 1** FL with MAPA

**Initialize:** Global model $W_0 \in \mathbb{R}^{d \times 1}$, reconstruction matrix $A_0 \in \mathbb{R}^{d \times p}$, projection matrix $\bar{B}_0 \leftarrow \mathbf{0} \in \mathbb{R}^{p \times 1}$, seed $r_0$

**for** each communication round $t = 1, \ldots, T - 1$ **do**
   **Server:** Broadcast global $\bar{B}_{t-1}$ and $r_{t-1}$
   **for** each client $i = 1, \ldots, N$ **in parallel do**
      **Client:** Receive $\bar{B}_{t-1}$ and $r_{t-1}$
      Update $W_t = W_{t-1} + A_{t-1}\bar{B}_{t-1}$
      Update $A_t = \mathcal{N}(0, \sigma)|r_{t-1}$
      Initialize $B_t^i \leftarrow \mathbf{0} \in \mathbb{R}^{p \times 1}$
      **for** each local epoch $e = 1, \ldots, E$ **do**:
         $\nabla B_t^i = \nabla_{B_t^i} \mathcal{L}(W_t + A_t B_t^i, \mathcal{D}_i)$
         Update $\hat{B}_t^i \leftarrow B_t^i - \eta \nabla B_t^i$
      **end for**
      Send updated $\hat{B}_t^i$ to server
   **end for**
   **Server:** Aggregate $\bar{B}_t \leftarrow \frac{1}{S}\sum_{i=1}^N b_i \hat{B}_t^i$
   Update global model $W_{t+1} \leftarrow W_t + A_t \bar{B}_t$
   update random seed $r_t$
**end for**
**Return:** Final global model $W_T$

---

This step focuses on finding the optimized projection vector $\hat{B}_t^i$ that minimizes the local loss function $\mathcal{L}(W_t + A_t B_t^i, \mathcal{D}_i)$, given the random matrix $A_t$. Here, the model weights are derived as $W_t + A_t B_t^i$, and $\mathcal{D}_i$ denotes client $i$-th local dataset. At each communication round $t \geq 1$, after initializing $A_t$ and $B_t^i$, clients perform local training to optimize $B_t^i$ using their local data $\mathcal{D}_i$. The gradient of the projection vector is computed as:

$$\nabla B_t^i = \nabla_{B_t^i} \mathcal{L}(W_t + A_t B_t^i, \mathcal{D}_i). \tag{2}$$

The optimized projection vector $\hat{B}_t^i$ is then updated using gradient descent:

$$\hat{B}_t^i \leftarrow B_t^i - \eta \nabla B_t^i, \tag{3}$$

where $\eta$ denotes the learning rate. After optimization, clients send their optimized projection vector $\hat{B}_t^i$ to the server. The low dimensionality of $\hat{B}_t^i$ compared to $W_t$ results in communication efficiency.

**Server-Side Aggregation and Global Model Update.** Upon receiving the projection vectors $\hat{B}_t^i$ and their corresponding weights $b_i$ (e.g., batch sizes or number of local samples) from the clients, the server aggregates them to form the global projection vector:

$$\bar{B}_t = \frac{\sum_{i=1}^N b_i \hat{B}_t^i}{\sum_{i=1}^N b_i}. \tag{4}$$

This weighted averaging captures the collective contribution of all clients, proportional to their data sizes. The server then broadcasts the aggregated projection vector $\bar{B}_t$ to all clients. After receiving $\bar{B}_t$, the server and all clients update their local models using the reconstruction matrix $A_t$ and the aggregated projection vector $\bar{B}_t$ as:

$$W_{t+1} = W_t + A_t \bar{B}_t. \tag{5}$$

This update integrates the clients' optimized directions into their local models and ensures synchronization across the network. This process repeats until the global model converges. Abbreviated pseudo-code is provided in Algorithm 1, while Appendix A offers a more detailed version.

## 4 CONVERGENCE ANALYSIS

We look into the convergence dynamics of FL with MAPA.

**Assumption 2.** *For each $i$, $\mathcal{L}_i(v)$ is $\beta$-smooth, i.e., $\|\nabla\mathcal{L}_i(u) - \nabla\mathcal{L}_i(v)\| \leq \beta\|u - v\|$ for any $u$, $v$.*

**Assumption 3.** *Variance of the stochastic gradient of $D_i$ is bounded for each client $i$, i.e.,* $\mathbb{E}\left[\left\|\nabla\mathcal{L}_i(W) - \widetilde{\nabla}\mathcal{L}_i(W)\right\|^2\right] \leq \sigma_l^2$.

**Theorem 2.** *Given a decreasing learning rate $\eta_t \leq \frac{1-4\epsilon}{4\beta(1+\epsilon)}$, the algorithm has the following convergence bound:*

$$\frac{1}{4H_T}\sum_{t=0}^{T-1}\eta_t\mathbb{E}\left[\|\nabla\mathcal{L}(W_t)\|^2\right] \leq \frac{\mathbb{E}\left[\mathcal{L}(W_0)\right] - \mathcal{L}^*}{H_T} + 2(\epsilon + \beta + \beta\epsilon)\sigma_l^2\left(\frac{1}{H_T}\sum_{t=0}^{T-1}\eta_t^2\right)$$

*where $H_T = \sum_{t=0}^{T-1}\eta_t$, $\epsilon$ is the distortion parameter from the JL Lemma, and $\mathcal{L}^*$ represents the minimum value of $\mathcal{L}(W)$.*

The proof can be found in Section D of the Appendix. With a decreasing learning rate, as $T \to \infty$, the upper bound converges to 0, confirming the convergence to a stationary point. As shown above, the convergence bound of MAPA is influenced by the $(3 - 2\rho)$ term, and we can see that the bound becomes the tightest and achieves the highest communication efficiency when there is no reconstruction error, i.e., when $\rho = 1$.

## 5 EXPERIMENTS

MAPA's effectiveness is assessed on image classification datasets: FMNIST Xiao et al. (2017), MNIST LeCun et al. (1998), CIFAR-10 and CIFAR-100 Krizhevsky et al. (2009). MNIST and FMNIST contain 60,000 training samples and 10,000 test samples, whereas CIFAR-10 and CIFAR-100 each comprise 50,000 training samples and 10,000 test samples. Unlike other tasks, CIFAR-100 includes 100 classes, each containing 500 training and 100 test samples. We employ a CNN model with 11k parameters for the MNIST and FMNIST datasets, a more substantial model with 1.4M parameters for CIFAR-10, and a ResNet model with 5.5 parameters for CIFAR-100.

We distribute the training set of each dataset among clients for model training, and the performance of the final global model is evaluated using the original test set. Our experimental setup contains $N = 100$ clients with non-IID data distribution. The non-IID distribution is created by splitting class data into 20 shards and then randomly assigning 5 shards from all class shards to each client by finding a permutation that uses the whole dataset while assigning two to five classes for each client. Similarly, for CIFAR-100, we attain twenty to fifty classes for each client.

Our MAPA framework is built using JAX (Bradbury et al., 2018), which facilitates extensive parallelization and, in particular, consistent random number generation across a large number of nodes and is designed for decoupled model parameters and architectures that ease the implementation of MAPA for factorization of parameters independent of the model architecture. MAPA is configured with 128 trainable parameters for MNIST and FMNIST while using 1024 for CIFAR-10 and CIFAR-100 and trains over 500 global rounds. We compare the performance of the proposed MAPA with FedAvg, FedAvg with Sparsification (Sparse), FedAvg with Quantization (Quant), EvoFed (Rahimi et al., 2024), as a SOTA baseline from compression techniques, Freeze A LoRA (FA-LoRA) inspired by Sun et al. (2024); Zhang et al. (2023); Zhu et al. (2024); Hao et al. (2024), as a SOTA baseline for factorization methods. In each scenario, we keep the same amount of trainable parameters.

**Results and Discussions.** We discuss the experimental results in detail and provide further insights into the performance of MAPA. The accuracy of MAPA, compared with multiple baseline methods and different datasets, is shown in Figure 5 (top row). MAPA's superior reconstruction outperforms all other methods in all tasks and delivers results comparable to FedAvg, utilizing a much smaller number of trainable parameters. Figure 5 (bottom-row) shows each method's minimum amount of

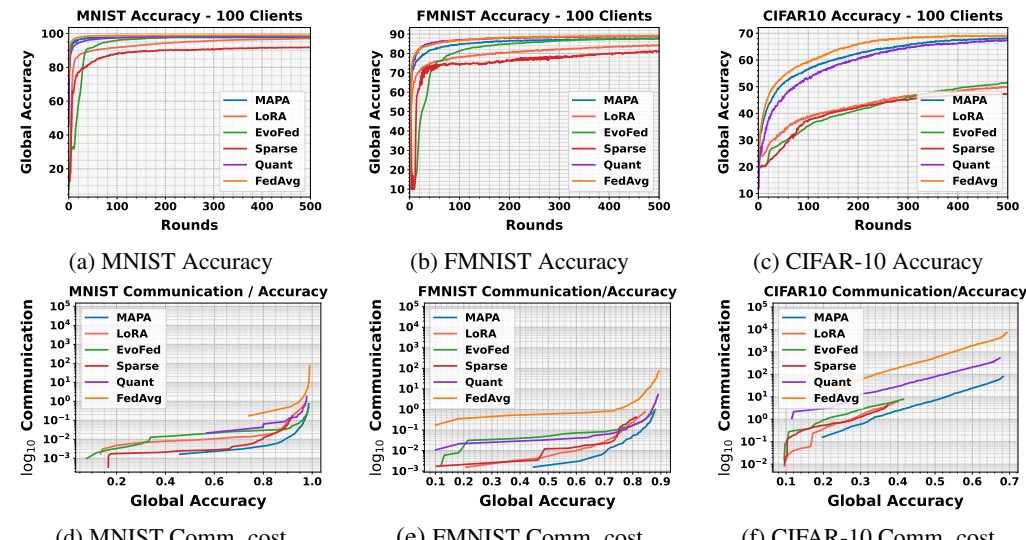

Figure 5: **Performance comparison** of MAPA and baseline methods on MNIST, FMNIST, and CIFAR-10 datasets. The top row shows the accuracy achieved by each method on the respective datasets, while the bottom row illustrates the communication cost associated with each method.

| | MNIST | | | FMNIST | | | CIFAR-10 | | | CIFAR-100 | | |
|---|---|---|---|---|---|---|---|---|---|---|---|---|
| **Methods** | Com. Cost | Local Acc. | Global Acc. | Com. Cost | Local Acc. | Global Acc. | Com. Cost | Local Acc. | Global Acc. | Com. Cost | Local Acc. | Global Acc. |
| FedAvg | 100% | 99.6% | 98.9% | 100% | 92.7% | 89.2% | 100% | 98.8% | 69.0% | 100% | 42.1% | 18.0% |
| Sparse | 15.3% | 97.7% | 93.8% | 24.1% | 84.4% | 81.1% | **1.0%** | 63.1% | 47.1% | 7.5% | 35.8% | 12.1% |
| Quantize | 31.3% | 98.8% | 97.6% | 24.1% | 83.6% | 81.1% | 5.0% | 84.8% | 67.1% | 54.2% | 32.1% | 10.2% |
| EvoFed | 9.4% | 98.6% | **98.5%** | 7.6% | 90.4% | 87.7% | 1.9% | 65.9% | 48.9% | 0.2% | 36.3% | 16.5% |
| FA-LoRA | 30.2% | 97.4% | 97.0% | 17.9% | 87.9% | 84.1% | 1.1% | 69.0% | 49.2% | 0.2% | 34.7% | 14.1% |
| MAPAX$_{d/64}$ | 3.1% | 98.8% | 98.1% | 3.4% | 91.0% | 87.7% | 1.1% | 88.7% | 68.2% | 0.1% | 36.6% | **16.8%** |
| MAPAX$_{d/256}$ | 3.0% | 98.8% | 98.2% | 3.3% | 91.3% | 87.9% | **1.0%** | 88.8% | 68.2% | 0.09% | 36.6% | **16.8%** |
| MAPAX$_{d/1024}$ | **2.9%** | 98.9% | 98.5% | **3.1%** | 91.2% | 87.9% | **1.0%** | 88.8% | 68.2% | **0.08%** | 36.7% | **16.8%** |
| **MAPA** | **2.9%** | **98.9%** | **98.5%** | **3.1%** | **91.4%** | **88.0%** | **1.0%** | **88.9%** | **68.3%** | **0.08%** | **36.7%** | **16.8%** |

Table 1: Performance of different methods presented in tabular form, corresponding to Figure 5.

communication to reach a certain accuracy. It can be seen that MAPA tends to utilize significantly less communication than other techniques, as the communication cost (y-axis) is in the $\log_{10}$ scale.

Table 1 summarizes each method's communication efficiency and performance on MNIST, FMNIST, CIFAR-10, and CIFAR-100 datasets. It presents maximum accuracy and the communication cost percentage compared to FedAvg. MAPA achieves significantly lower communication costs than FedAvg while maintaining competitive accuracy levels. In MNIST and FMNIST datasets, MAPA achieves 98.5% and 98.6% of FedAvg accuracy while having only 3% of FedAvg communication. Similarly, in CIFAR-10 and CIFAR-100 datasets, it reaches 98.9% and 93.3% of FedAvg accuracy with around 1.0% of FedAvg communication.

**Additional Results.** Additional results regarding IID distribution and client sampling have been provided in Appendix F, while Appendix B provides empirical results regarding the matrix $A$ initialization, and the hyperparameters and network architecture details are provided in Appendix G.

**MAPAX Performance.** We provide additional results to evaluate MAPAX factorizations regarding their performance and communication efficiency. Figure 6 first row visualizes our theoretical finding, providing a map to navigate through factorization space regarding our MAPAX, and the second row presents our empirical experiments results of MAPAX accuracy for each factorization. For clarity, the Axes of Figure 6 and values for communication and memory amount are presented in $\log_2$ scale and denote discrete values representing matrix dimensions.

Figure 6 (a) visualizes efficient, same, over communication zones. The factorization region that achieves communication efficiency by having fewer trainable parameters (smaller $B$), shown in blue, and the over-communication zone in red, denotes the factorization with more trainable parameters (larger $B$) than the original model. This figure shows that MAPAX can cover possible factorization in the communication-efficient zone by selecting $k$ and $p$ accordingly.

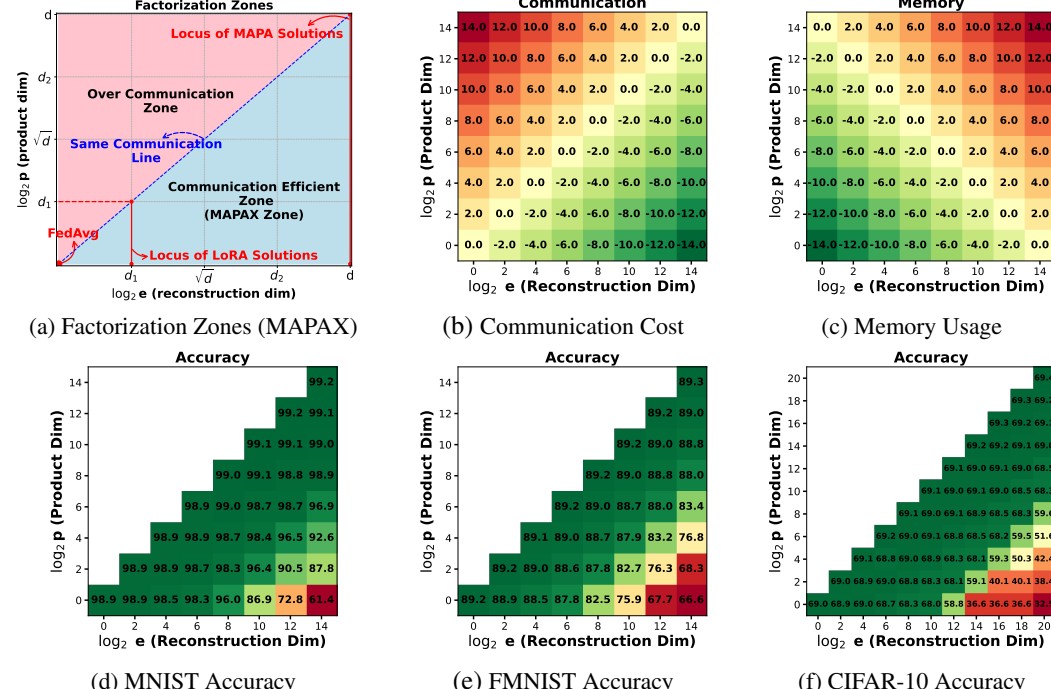

Figure 6: Each point on the plot denotes different factorization with various sizes of reconstruction matrix and projection vector. **First row** presents the theoretical findings: (a) denotes the zones regarding communication efficiency, while (b) and (c) show a numerical example of communication burden and memory usage. **Second row** provides results of each factorization accuracy on MNIST, FMNIST, and CIFAR-10 datasets.

Figure 6 (b) shows each factorization's communication coefficient in $\log_2$ scale. Given a model with $d$ parameters, the communication cost will be $d2^c$, where $c$ is the communication coefficient according to the table. Similarly, Figure 6 (c) shows each factorization's memory usage coefficient in $\log_2$ scale. Given a model with $d$ parameters, the memory usage will be $d2^m$, where $m$ is the memory usage coefficient according to the table.

Figure 6 (bottom row) shows the results of empirical experiments of possible factorization on MNIST, FMNIST, and CIFAR-10 with non-iid distribution on 100 clients. We conducted 28 factorizations for MNIST and FMNIST and 55 for CIFAR-10 by adjusting $p$ and $e$ values. It is evident among all tasks and confirming our theoretical analysis in Appendix E that the model accuracy is highly correlated with communication cost. As Figure 6 (bottom row) shows, the performance drops suddenly as communication goes to zero, while it will saturate after having an adequate amount of communication, which suggests the existence of a low-rank structure for the neural network gradient. As shown in Figure 6, we can have varying degrees of memory efficiency for a given communication rate. Therefore, MAPAX takes advantage of this fact and provides memory and computationally efficient solutions for slightly underperforming MAPA, saving quadratic order of memory and computation. Appendix E demonstrates a complete analysis regarding MAPAX and how parameters can be tuned considering the task's requirements.

## 6 CONCLUSION

We introduced Model-Agnostic Projection Adaptation (MAPA), a novel technique for enhancing communication efficiency in FL by treating the entire model parameter space as a single vector and factorizing it into a fixed reconstruction matrix and a trainable projection vector. This approach can utilize a new reconstruction matrix at each round, increasing the expressiveness of model updates without additional communication costs. Our theoretical analysis established the convergence of MAPA and demonstrated and provided MAPAX extension as a solution for high memory consumption. Extensive experiments on different datasets showed that MAPA provides a significant advancement in improving the efficiency and practicality of FL, offering a promising direction for future research in distributed machine learning.

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

## A   FULL PSEUDOCODE FOR FEDERATED LEARNING WITH MAPA

---

**Algorithm 2** FL with MAPA

---

1: **Initialization:**
2: - Initialize global model $W_0 \in \mathbb{R}^{d \times 1}$ with random seed $r_0$ in all clients and server
3: - Initialize reconstruction matrix $A_0 \in \mathbb{R}^{d \times p}$ with random vectors
4: - Set initial global projection matrix $\bar{B}_0 \leftarrow \mathbf{0} \in \mathbb{R}^{p \times 1}$
5: **for** each communication round $t = 1, \ldots, T - 1$ **do**
6:     **Server-Side:**
7:     - Broadcast global projection matrix $\bar{B}_{t-1}$ and PRNG seed $r_{t-1}$
8:     **for** each client $i = 1, \ldots, N$ **in parallel do**
9:         **Client-Side:**
10:         - Receive $\bar{B}_{t-1}$ and $r_{t-1}$
11:         - Update the local model by $W_t = W_{t-1} + A_{t-1}\bar{B}_{t-1}$
12:         - Update reconstruction matrix $A_t = \mathcal{N}(0, \sigma)|r_{t-1}$
13:         - Initialize local projection matrix $B_t^i \leftarrow \mathbf{0} \in \mathbb{R}^{p \times 1}$
14:         **for** each local epoch $e = 1, \ldots, E$ **do:**
15:             - Compute gradient of loss w.r.t $B_t^i$:

$$\nabla B_t^i = \nabla_{B_t^i} \mathcal{L}(W_t + A_t B_t^i, \mathcal{D}_i)$$

16:             - Update $B_t^i$ using optimizer (e.g., SGD, Adam):

$$\hat{B}_t^i \leftarrow B_t^i - \eta \nabla B_t^i$$

17:         **end for**
18:         - Send updated $B_t^i$ to server
19:     **end for**
20:     **Server-Side:**
21:     - Update reconstruction matrix $A_t = \mathcal{N}(0, \sigma)|r_{t-1}$
22:     - Aggregate projection matrix:

$$\bar{B}_t \leftarrow \frac{1}{S} \sum_{i=1}^{N} b_i \hat{B}_t^i$$

23:     - Update global model: $W_{t+1} \leftarrow W_t + A_t \bar{B}_t$
24:     - Generate a new random seed $r_t$ given previous seed $r_{t-1}$
25: **end for**
26: **Return:** Final global model $W_T$

---

## B  FRESH INITIALIZATION OF RECONSTRUCTION MATRIX $A$

The common practice of implementing matrix factorization in communication-efficient FL involves using a fixed and frozen reconstruction matrix throughout the whole training. In contrast, we found that having a reconstructed matrix generated fresh and independently each round outperforms this traditional choice without any additional communication overhead. Figure 7 shows the evidence of this improvement in the case of FMNIST training with 100 clients.

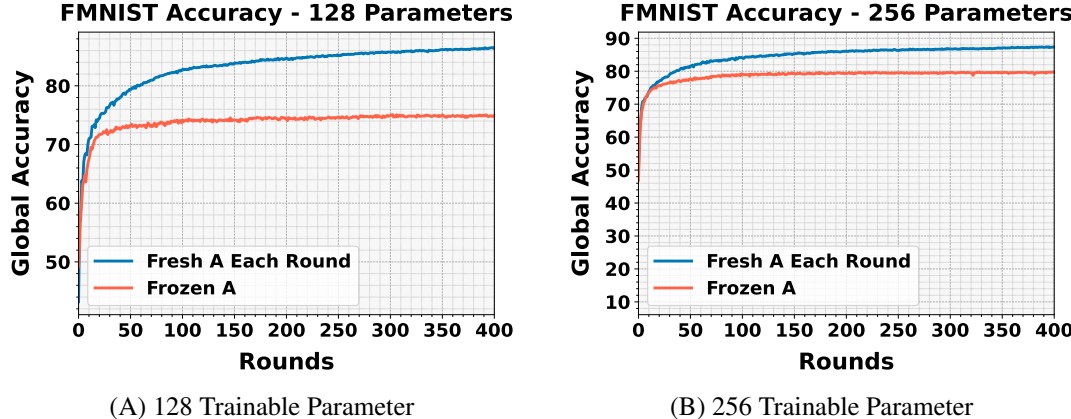

(A) 128 Trainable Parameter        (B) 256 Trainable Parameter

Figure 7: Comparison of having a Fresh and Frozen reconstruction matrix at each round. It shows that the Fresh reconstruction matrix outperforms the Frozen as it has stronger exploration and more chance to escape local minima.

## C  PROOF OF PROPOSITIONS

### C.1  DEFINITION 2: REPRESENTATION CERTAINTY

**Definition 2** (**Representation Certainty**). *Using the same factorization as in Definition 1. The* ***representation certainty*** *is defined as the inverse of the error rate variance. The error rate measures the expected error of the factorization to represent the original matrix, given a full-rank matrix A (Assumption 1). The error expectation and variance are defined as:*

$$\mathbb{E}_A \left[ \|W - AB\|_2^2 \right] = \left( 1 - \frac{q}{d_1} \right), \mathbf{Var}_A \left[ \|W - AB\|_2^2 \right] = \left( \frac{2q(d_1 - q)}{d_1^2(d_1 + 2)} \right)$$

*Therefore, given a constant communication overhead and error expectation $\frac{d_1}{q} = r$ we have:*

$$\mathrm{RC}(\Delta W, \mathcal{F}) = \frac{1}{\mathbf{Var}_A[\mathbf{E}]} = \frac{r^3 q + r^2}{2(r - 1)} \propto q$$

Let $x \in \mathbb{R}^d$, and let $A \in \mathbb{R}^{d \times p}$ be a random Gaussian matrix. The projection of $x$ onto the subspace spanned by $A$ is $P_A x$. The error rate $E$ is defined as:

$$E = \frac{\|x - P_A x\|_2^2}{\|x\|_2^2}.$$

Using the Pythagorean theorem:

$$\|x\|_2^2 = \|P_A x\|_2^2 + \|x - P_A x\|_2^2,$$

we rewrite $E$ as:

$$E = \frac{\|x\|_2^2 - \|P_A x\|_2^2}{\|x\|_2^2} = 1 - \frac{\|P_A x\|_2^2}{\|x\|_2^2}.$$

The expected value of $\|P_A x\|_2^2$ for a random Gaussian projection is:

$$\mathbb{E}[\|P_A x\|_2^2] = \frac{p}{d}\|x\|_2^2.$$

Substituting this into $E$:

$$\mathbb{E}[E] = 1 - \frac{\mathbb{E}[\|P_A x\|_2^2]}{\|x\|_2^2} = 1 - \frac{\frac{p}{d}\|x\|_2^2}{\|x\|_2^2} = 1 - \frac{p}{d}.$$

Thus:

$$\mathbb{E}[E] = 1 - \frac{p}{d}.$$

The variance of $E$ is:

$$\mathrm{Var}(E) = \frac{\mathrm{Var}(\|P_A x\|_2^2)}{\|x\|_2^4}.$$

For a random Gaussian projection:

$$\mathrm{Var}(\|P_A x\|_2^2) = \mathbb{E}[\|P_A x\|_2^4] - \left( \mathbb{E}[\|P_A x\|_2^2] \right)^2.$$

The moments are:

$$\mathbb{E}[\|P_A x\|_2^4] = \frac{p(p + 2)}{d(d + 2)}\|x\|_2^4, \quad \left( \mathbb{E}[\|P_A x\|_2^2] \right)^2 = \frac{p^2}{d^2}\|x\|_2^4.$$

Thus:

$$\mathrm{Var}(\|P_A x\|_2^2) = \frac{p(p + 2)}{d(d + 2)}\|x\|_2^4 - \frac{p^2}{d^2}\|x\|_2^4 = \frac{2p(d - p)}{d^2(d + 2)}\|x\|_2^4.$$

Substituting into $\mathrm{Var}(E)$:

$$\mathrm{Var}(E) = \frac{\frac{2p(d-p)}{d^2(d+2)}\|x\|_2^4}{\|x\|_2^4} = \frac{2p(d - p)}{d^2(d + 2)}.$$

The expected value and variance of the error rate are:

$$\mathbb{E}[E] = 1 - \frac{p}{d}, \quad \mathrm{Var}(E) = \frac{2p(d - p)}{d^2(d + 2)}.$$

## C.2 Proposition 1: Relaxed Low-Rank Factorization Superiority

**Proposition 1** (**Relaxed Low-Rank Factorization Superiority**). *Let $\Delta W \in \mathbb{R}^{d_1 \times d_2}$ be the update matrix of one layer, factorized in low-rank as $\Delta W = AB$, where $A \in \mathbb{R}^{d_1 \times q}$ is a shared random matrix and $B \in \mathbb{R}^{q \times d_2}$ is the trainable matrix, with $q \leq \min(d_1, d_2)$ being the factorization rank, By reshaping $\Delta W$ into $\Delta W' \in \mathbb{R}^{(d_1 d_2)/k \times k}$ for some integer $k < d_2$, the factorization of $\Delta W'$ can achieve a higher **representation certainty** while requiring same **communication overhead** compared to the conventional low-rank factorization of $\Delta W$.*

*Proof.* Let $\Delta W \in \mathbb{R}^{d_1 \times d_2}$ represent the update matrix, which is conventionally factorized as $\Delta W = AB$, where $A \in \mathbb{R}^{d_1 \times q}$ is a fixed random reconstruction matrix, and $B \in \mathbb{R}^{q \times d_2}$ is the trainable projection matrix. Here, $q \leq \min(d_1, d_2)$ denotes the factorization rank.

According to Definitions 1 and 2, the communication overhead (CO) and representation certainty (RC) for the conventional low-rank factorization are expressed as:

$$\text{CO} = \frac{q}{d_1}, \quad \text{RC} = \frac{d_1^2(d_1 + 2)}{2q(d_1 - q)}.$$

Now, consider reshaping $\Delta W$ into $\Delta W' \in \mathbb{R}^{(d_1 d_2)/k \times k}$ for some integer $k < d_2$ that divides $d_1 d_2$. Factorize $\Delta W'$ as $\Delta W' = A'B'$, where $A' \in \mathbb{R}^{(d_1 d_2)/k \times p}$ is a fixed random reconstruction matrix and $B' \in \mathbb{R}^{p \times k}$ is the trainable projection matrix. Following Definitions 1 and 2, the communication overhead (CO′) and representation certainty (RC′) for this relaxed low-rank factorization are given by:

$$\text{CO}' = \frac{pk}{d_1 d_2}, \quad \text{RC}' = \frac{\left(\frac{d_1 d_2}{k}\right)^2 \left(\frac{d_1 d_2}{k} + 2\right)}{2p\left(\frac{d_1 d_2}{k} - p\right)}.$$

Assuming the communication overhead is the same in both cases, i.e., $\frac{pk}{d_1 d_2} = \frac{q}{d_1} = \frac{1}{r}$, it follows that:

$$\text{RC} = \frac{r^3 q + r^2}{2(r-1)}, \quad \text{RC}' = \frac{r^3 p + r^2}{2(r-1)}.$$

Furthermore, given $k < d_2$, we derive:

$$\frac{pk}{d_1 d_2} = \frac{q}{d_1} \implies \frac{pk}{d_2} = q \implies p = \frac{d_2}{k}q \implies p > q \implies \text{RC}' > \text{RC}.$$

In conclusion, by reshaping $\Delta W \in \mathbb{R}^{d_1 \times d_2}$ into $\Delta W' \in \mathbb{R}^{(d_1 d_2)/k \times k}$ with $k \leq d_2$, one can select a rank $p = \frac{d_2}{k}q$, thereby achieving a higher representation certainty while maintaining the same communication overhead. This establishes the superiority of the relaxed low-rank factorization under the given conditions. $\square$

### C.3 PROPOSITION 2: LAYER-INDEPENDENT FACTORIZATION SUPERIORITY

**Proposition 2 (Layer-Independent Factorization Superiority).** *Let $\Delta W_i \in \mathbb{R}^{d_1^i \times d_2^i}$ be the update matrix of the $i$-th layer of a model, and let $\Delta W_i' \in \mathbb{R}^{d_1^i d_2^i \times 1}$ be its reshaped single-vector form. In single-vector factorization methods, $\Delta W_i'$ is factorized as $\Delta W_i' = A_i B_i$, where $A_i \in \mathbb{R}^{d_1^i d_2^i \times q_i}$ and $B_i \in \mathbb{R}^{q_i \times 1}$, with $q_i \leq d_1^i d_2^i$. By concatenating the reshaped weights $\Delta W_i'$ into $\Delta W' \in \mathbb{R}^{d \times 1}$, where $d = \sum_{i=1}^{n} d_1^i d_2^i$. The factorization of $\Delta W'$ can achieve a higher **representation certainty** while requiring the same **communication overhead** than the conventional single-vector factorization methods applied separately to each layer.*

*Proof.* Let $\Delta W_i \in \mathbb{R}^{d_1^i \times d_2^i}$ be the update matrix of the $i$-th layer, which we reshape into its single-vector form $\Delta W_i' \in \mathbb{R}^{d_1^i d_2^i \times 1}$. In conventional single-vector factorization methods applied separately to each layer, $\Delta W_i'$ is factorized as:

$$\Delta W_i' = A_i B_i,$$

where $A_i \in \mathbb{R}^{d_1^i d_2^i \times q_i}$ is a fixed random reconstruction matrix, and $B_i \in \mathbb{R}^{q_i \times 1}$ is the trainable projection matrix, with $q_i \leq d_1^i d_2^i$.

According to Definitions 1 and 2, the *communication overhead* and *representation certainty* are given by:

$$\text{CO} = \frac{\sum_{i=1}^{n} q_i}{\sum_{i=1}^{n} d_1^i d_2^i} \quad ; \quad \text{RC}_i = \frac{(d_1^i d_2^i)^2 (d_1^i d_2^i + 2)}{2 q_i (d_1^i d_2^i - q_i)}.$$

Now, consider concatenating the reshaped vectors $\Delta W_i'$ from all $n$ layers into a single vector $\Delta W' \in \mathbb{R}^{d \times 1}$, where:

$$d = \sum_{i=1}^{n} d_1^i d_2^i.$$

We factorize the concatenated vector $\Delta W'$ as:

$$\Delta W' = AB,$$

where $A \in \mathbb{R}^{d \times q}$ is a fixed random reconstruction matrix, and $B \in \mathbb{R}^{q \times 1}$ is the trainable projection matrix, with $q \leq d$.

The *communication overhead* and *representation certainty* for the concatenated factorization are:

$$\text{CO}' = \frac{q}{d} \quad ; \quad \text{RC}' = \frac{d^2 (d + 2)}{2 q (d - q)}.$$

Assuming the communication overhead is identical for both methods for some $r \geq 1$, we have:

$$\frac{q_i}{d_1^i d_2^i} = \frac{q}{d} = \frac{1}{r} \implies q = \frac{d}{d_1^i d_2^i} q_i.$$

Substituting $d, d_1^i d_2^i$ into the expression for $\text{RC}_i$ and $\text{RC}'$:

$$\text{RC}_i = \frac{r^3 q_i + r^2}{2(r-1)}, \quad \text{RC}' = \frac{r^3 q + r^2}{2(r-1)}.$$

Furthermore, given $d_1^i d_2^i < d$, we derive:

$$q = \frac{d}{d_1^i d_2^i} q_i \implies q > q_i \implies \text{RC}' > \text{RC}_i.$$

In conclusion, by concatenating the reshaped weights $\Delta W_i' \in \mathbb{R}^{d_1^i d_2^i \times 1}$ to $W' \in \mathbb{R}^{d \times 1}$, where $i \geq 1$ we can select the rank $q = \frac{d}{d_i} q_i$ to achieve a higher representation certainty while reducing the communication overhead compared to the conventional single-vector factorization methods applied separately to each layer under given conditions. $\square$

## C.4   Proposition 3: MAPAX Generalization

**Proposition 3 (MAPAX Generalization).** *Let $\Delta W_i \in \mathbb{R}^{d_1^i \times d_2^i}$ be the update matrix of the $i$-th layer of a model, and let $\Delta W = \text{vec}(\Delta W_1, \Delta W_2, \ldots, \Delta W_n) \in \mathbb{R}^d$ be the vectorization (concatenation) of all $\Delta W_i$, where $d = \sum_{i=1}^n d_1^i d_2^i$. In comparison to MAPA, $\text{MAPAX}_k$ factorization allocates $k^2$ times less memory for the same **communication overhead** and **error rate**, for the cost of $k$ times worse **representation certainty**, in other words, more $k$ times more error rate variance.*

*Proof.* Let $\Delta W \in \mathbb{R}^{d \times 1}$ be the update matrix. In MAPA factorization $\Delta W$ is factorized as:

$$\Delta W = AB,$$

where $A \in \mathbb{R}^{d \times p}$ is a fixed random reconstruction matrix, and $B \in \mathbb{R}^{p \times 1}$ is the trainable projection matrix.

According to Definitions 1 and 2, the *communication overhead* and *representation certainty* are given by:

$$\text{CO} = \frac{p}{d} \quad ; \quad \text{RC} = \frac{d^2(d+2)}{2p(d-p)}.$$

Now, consider $\text{MAPAX}_k$ reshapes update vector to $\Delta W' \in \mathbb{R}^{\frac{d}{k} \times k}$, and factorizes $\Delta W'$ as:

$$\Delta W' = A'B',$$

where $A' \in \mathbb{R}^{\frac{d}{k} \times q}$ is a fixed random reconstruction matrix, and $B' \in \mathbb{R}^{q \times k}$ is the trainable projection matrix.

The *communication overhead* and *representation certainty* for the concatenated factorization are:

$$\text{CO}' = \frac{qk}{d} \quad ; \quad \text{RC}' = \frac{(\frac{d}{k})^2((\frac{d}{k})+2)}{2q((\frac{d}{k})-q)}.$$

Assuming the communication overhead is identical for both methods for some $r \geq 1$, we have:

$$\frac{qk}{d} = \frac{p}{d} = \frac{1}{r} \implies q = \frac{p}{k}.$$

Substituting $d$ into the expression for RC and $\text{RC}'$:

$$\text{RC} = \frac{r^3 p + r^2}{2(r-1)}, \quad \text{RC}' = \frac{r^3 \frac{p}{k} + r^2}{2(r-1)}.$$

Therefore, $\text{MAPAX}_k$ has $k$ times less representation certainty compared to MAPA.

On the other hand, the memory allocation of matrix $A$ and $A'$ can be computed as:

$$\text{Size}(A) = dp \quad ; \quad \text{Size}(A') = \frac{dp}{k^2},$$

demonstrating that $\text{MAPAX}_k$ utilizes $k^2$ times less memory compared to MAPA. $\qquad\square$

# D    PROOF OF THEOREM

## D.1    ASSUMPTIONS AND PRELIMINARIES

We restate the key assumptions required for the convergence analysis.

**Assumption 1** (Smoothness). *For each $i$, $\mathcal{L}_i(W)$ is $\beta$-smooth, i.e.,*

$$\|\nabla\mathcal{L}_i(u) - \nabla\mathcal{L}_i(v)\| \le \beta\|u - v\|, \quad \text{for all } u, v.$$

**Assumption 2** (Bounded Variance of Stochastic Gradients). *The variance of the stochastic gradient estimator $\widetilde{\nabla}\mathcal{L}_i(W_t)$ is bounded, i.e., $\mathbb{E}\left[\left\|\widetilde{\nabla}\mathcal{L}_i(W_t) - \nabla\mathcal{L}_i(W_t)\right\|^2\right] \le \sigma_l^2$, for all clients $i$ and iterations $t$.*

**Lemma 1** (Johnson-Lindenstrauss Lemma). *Given $0 < \epsilon < 1$, a set of points $\{x_1, x_2, \ldots, x_N\} \subset \mathbb{R}^d$, and a target dimension $k = O\left(\frac{\log N}{\epsilon^2}\right)$, there exists a random linear mapping $P \in \mathbb{R}^{k \times d}$ such that for all $i, j$:*

$$(1 - \epsilon)\|x_i - x_j\|^2 \le \|Px_i - Px_j\|^2 \le (1 + \epsilon)\|x_i - x_j\|^2.$$

In our context, the random projection matrices $B_t^i$ and reconstruction matrices $A_t$ satisfy the JL property with high probability.

## D.2    PROOF OF THEOREM 1

**Theorem 1.** *Given a decreasing learning rate $\eta_t \le \frac{1 - 4\epsilon}{4\beta(1 + \epsilon)}$, the algorithm has the following convergence bound:*

$$\frac{1}{4H_T}\sum_{t=0}^{T-1}\eta_t\mathbb{E}\left[\|\nabla\mathcal{L}(W_t)\|^2\right] \le \frac{\mathbb{E}[\mathcal{L}(W_0)] - \mathcal{L}^*}{H_T} + 2(\epsilon + \beta + \beta\epsilon)\sigma_l^2\left(\frac{1}{H_T}\sum_{t=0}^{T-1}\eta_t^2\right)$$

*where $H_T = \sum_{t=0}^{T-1}\eta_t$, $\epsilon$ is the distortion parameter from the JL Lemma, and $\mathcal{L}^*$ represents the minimum value of $\mathcal{L}(W)$.*

*Proof.* By the $\beta$-smoothness of $\mathcal{L}(W)$ and taking expectation on both sides, we have

$$\mathbb{E}[\mathcal{L}(W_{t+1}) - \mathcal{L}(W_t)] \le \mathbb{E}[\langle\nabla\mathcal{L}(W_t), W_{t+1} - W_t\rangle] + \frac{\beta}{2}\mathbb{E}\left[\|W_{t+1} - W_t\|^2\right]. \quad (6)$$

Using the update rule $W_{t+1} = W_t - \eta_t A_t\bar{B}_t$, where $\bar{B}_t = \frac{1}{N}\sum_{i=1}^{N}B_t^i$, we can rewrite the first term as:

$$\mathbb{E}[\langle\nabla\mathcal{L}(W_t), W_{t+1} - W_t\rangle] = -\eta_t\mathbb{E}\left[\langle\nabla\mathcal{L}(W_t), A_t\bar{B}_t\rangle\right]$$

$$= -\eta_t\mathbb{E}\left[\left\langle\nabla\mathcal{L}(W_t), A_t\left(\frac{1}{N}\sum_{i=1}^{N}B_t^i\right)\right\rangle\right]$$

$$= -\eta_t\mathbb{E}\left[\left\langle\nabla\mathcal{L}(W_t), \frac{1}{N}\sum_{i=1}^{N}A_tB_t^i\right\rangle\right].$$

We decompose $A_tB_t^i$ as:

$$\widetilde{\nabla}\mathcal{L}_i(W_t) = A_tB_t^i + e_t^i,$$

where $e_t^i = A_tB_t^i - \widetilde{\nabla}\mathcal{L}_i(W_t)$ is the projection error.

Substituting back, we have:

$$\mathbb{E}[\langle\nabla\mathcal{L}(W_t), W_{t+1} - W_t\rangle] = -\eta_t\mathbb{E}\left[\left\langle\nabla\mathcal{L}(W_t), \frac{1}{N}\sum_{i=1}^{N}\left(\widetilde{\nabla}\mathcal{L}_i(W_t) - e_t^i\right)\right\rangle\right]$$

$$= \underbrace{-\eta_t\mathbb{E}\left[\left\langle\nabla\mathcal{L}(W_t), \frac{1}{N}\sum_{i=1}^{N}\widetilde{\nabla}\mathcal{L}_i(W_t)\right\rangle\right]}_{A_1} + \underbrace{\eta_t\mathbb{E}\left[\left\langle\nabla\mathcal{L}(W_t), \frac{1}{N}\sum_{i=1}^{N}e_t^i\right\rangle\right]}_{A_2}.$$

We will now concentrate on $A_1$ as:

$$A_1 = -\eta_t \mathbb{E}\left[\left\langle \nabla\mathcal{L}(W_t), \frac{1}{N}\sum_{i=1}^{N}\nabla\mathcal{L}_i(W_t)\right\rangle\right]$$

$$= -\frac{\eta_t}{N}\sum_{i=1}^{N}\mathbb{E}\left[\langle\nabla\mathcal{L}(W_t), \nabla\mathcal{L}_i(W_t)\rangle\right]$$

$$\overset{=}{_{(a)}} -\frac{\eta_t}{2N}\sum_{i=1}^{N}\left\{\mathbb{E}\left[\|\nabla\mathcal{L}(W_t)\|^2\right] + \mathbb{E}\left[\left\|\nabla\mathcal{L}_i(W_t)\right\|^2\right]\right\} + \frac{\eta_t}{2}\mathbb{E}\left[\left\|\underbrace{\nabla\mathcal{L}(W_t) - \frac{1}{N}\sum_{i=1}^{N}\nabla\mathcal{L}_i(W_t)}_{=0}\right\|^2\right]$$

$$= -\frac{\eta_t}{2}\mathbb{E}\left[\|\nabla\mathcal{L}(W_t)\|^2\right] - \frac{\eta_t}{2N}\sum_{i=1}^{N}\mathbb{E}\left[\left\|\nabla\mathcal{L}_i(W_t)\right\|^2\right]$$

where (a) uses $\langle a, b\rangle = \frac{1}{2}\{\|a\|^2 + \|b\|^2 - \|a-b\|^2\}$. We now turn our attention to $A_2$ as:

Next, we focus on $A_2$:

$$A_2 = \eta_t\mathbb{E}\left[\left\langle\nabla\mathcal{L}(W_t), \frac{1}{N}\sum_{i=1}^{N}e_t^i\right\rangle\right]$$

$$\overset{\leq}{_{(a)}} \frac{\eta_t}{4}\mathbb{E}\left[\|\nabla\mathcal{L}(W_t)\|^2\right] + \eta_t\mathbb{E}\left[\left\|\frac{1}{N}\sum_{i=1}^{N}e_t^i\right\|^2\right]$$

$$\overset{\leq}{_{(b)}} \frac{\eta_t}{4}\mathbb{E}\left[\|\nabla\mathcal{L}(W_t)\|^2\right] + \frac{\eta_t}{N}\mathbb{E}\left[\left\|\sum_{i=1}^{N}e_t^i\right\|^2\right]$$

$$\overset{\leq}{_{(c)}} \frac{\eta_t}{4}\mathbb{E}\left[\|\nabla\mathcal{L}(W_t)\|^2\right] + \frac{\epsilon\eta_t}{N}\mathbb{E}\left[\left\|\sum_{i=1}^{N}\widetilde{\nabla}\mathcal{L}_i(W_t)\right\|^2\right]$$

$$\overset{\leq}{_{(d)}} \frac{\eta_t}{4}\mathbb{E}\left[\|\nabla\mathcal{L}(W_t)\|^2\right] + \frac{2\epsilon\eta_t}{N}\sum_{i=1}^{N}\left\{\mathbb{E}\left[\|\nabla\mathcal{L}_i(W_t)\|^2\right] + \mathbb{E}\left[\left\|\widetilde{\nabla}L_i(W_t) - \nabla\mathcal{L}_i(W_t)\right\|^2\right]\right\}$$

$$\overset{\leq}{_{(e)}} \frac{\eta_t}{4}\mathbb{E}\left[\|\nabla\mathcal{L}(W_t)\|^2\right] + \frac{2\epsilon\eta_t}{N}\sum_{i=1}^{N}\mathbb{E}\left[\|\nabla\mathcal{L}_i(W_t)\|^2\right] + 2\epsilon\eta_t^2\sigma_l^2$$

where (a) uses $\langle a, b\rangle \leq \frac{1}{4}\|a\|^2 + \|b\|^2$, and (b) follows Jensen's inequality, (c) comes from JL Lemma, (d) follows the inequality $\|a + b\|^2 \leq 2\|a\|^2 + 2\|b\|^2$, and (e) is based on Assumption 2. On the other hand, we can also place a bound on the second term $\mathbb{E}\left[\|W_{t+1} - W_t\|^2\right]$ as shown below:

$$\mathbb{E}\left[\|W_{t+1} - W_t\|^2\right] = \mathbb{E}\left[\left\|\eta_t A_t \bar{B}_t\right\|^2\right] = \mathbb{E}\left[\left\|\eta_t A_t\left(\frac{1}{N}\sum_{i=1}^{N}B_t^i\right)\right\|^2\right]$$

$$\overset{\leq}{_{(a)}} 2\eta_t^2\mathbb{E}\left[\left\|\frac{1}{N}\sum_{i=1}^{N}\widetilde{\nabla}\mathcal{L}_i(W_t)\right\|^2\right] + 2\eta_t^2\mathbb{E}\left[\left\|\frac{1}{N}\sum_{i=1}^{N}\left\{A_t B_t^i - \widetilde{\nabla}\mathcal{L}_i(W_t)\right\}\right\|^2\right]$$

$$\underset{(b)}{\leq} \frac{2\eta_t^2}{N} \mathbb{E}\left[\left\|\sum_{i=1}^N \widetilde{\nabla}\mathcal{L}_i(W_t)\right\|^2\right] + \frac{2\eta_t^2}{N}\mathbb{E}\left[\left\|\sum_{i=1}^N \left\{A_t B_t^i - \widetilde{\nabla}\mathcal{L}_i(W_t)\right\}\right\|^2\right]$$

$$= \frac{2\eta_t^2}{N} \mathbb{E}\left[\left\|\sum_{i=1}^N \widetilde{\nabla}\mathcal{L}_i(W_t)\right\|^2\right] + \frac{2\eta_t^2}{N}\mathbb{E}\left[\left\|\sum_{i=1}^N e_t^i\right\|^2\right]$$

$$\underset{(c)}{\leq} \frac{4\eta_t^2}{N} \sum_{i=1}^N \left\{\mathbb{E}\left[\|\nabla\mathcal{L}_i(W_t)\|^2\right] + \mathbb{E}\left[\left\|\widetilde{\nabla}L_i(W_t) - \nabla\mathcal{L}_i(W_t)\right\|^2\right]\right\} + \frac{2\eta_t^2}{N}\mathbb{E}\left[\left\|\sum_{i=1}^N e_t^i\right\|^2\right]$$

$$\underset{(d)}{\leq} \frac{4\eta_t^2}{N} \sum_{i=1}^N \mathbb{E}\left[\|\nabla\mathcal{L}_i(W_t)\|^2\right] + \frac{2\eta_t^2}{N}\mathbb{E}\left[\left\|\sum_{i=1}^N e_t^i\right\|^2\right] + 4\eta_t^2\sigma_l^2$$

$$\underset{(e)}{\leq} \frac{4\eta_t^2}{N} \sum_{i=1}^N \mathbb{E}\left[\|\nabla\mathcal{L}_i(W_t)\|^2\right] + \frac{2\epsilon\eta_t^2}{N}\mathbb{E}\left[\left\|\sum_{i=1}^N \widetilde{\nabla}\mathcal{L}_i(W_t)\right\|^2\right] + 4\eta_t^2\sigma_l^2$$

$$\underset{(f)}{\leq} \frac{4\eta_t^2}{N} \sum_{i=1}^N \mathbb{E}\left[\|\nabla\mathcal{L}_i(W_t)\|^2\right] + \frac{4\epsilon\eta_t^2}{N}\sum_{i=1}^N\left\{\mathbb{E}\left[\|\nabla\mathcal{L}_i(W_t)\|^2\right] + \mathbb{E}\left[\left\|\widetilde{\nabla}L_i(W_t) - \nabla\mathcal{L}_i(W_t)\right\|^2\right]\right\} + 4\eta_t^2\sigma_l^2$$

$$\underset{(g)}{\leq} \frac{4\eta_t^2}{N} \sum_{i=1}^N \mathbb{E}\left[\|\nabla\mathcal{L}_i(W_t)\|^2\right] + \frac{4\epsilon\eta_t^2}{N}\sum_{i=1}^N\mathbb{E}\left[\|\nabla\mathcal{L}_i(W_t)\|^2\right] + 4\epsilon\eta_t^2\sigma_l^2 + 4\eta_t^2\sigma_l^2$$

$$= \frac{4(1+\epsilon)\eta_t^2}{N} \sum_{i=1}^N \mathbb{E}\left[\|\nabla\mathcal{L}_i(W_t)\|^2\right] + 4(1+\epsilon)\eta_t^2\sigma_l^2$$

where (a), (c), and (f) are based on the inequality $\|a+b\|^2 \leq 2\|a\|^2 + 2\|b\|^2$, (b) comes from Jensen's inequality, (d), (g) derive from Assumption 2, and (e) comes from JL Lemma.

By utilizing the previously established bounds for $\mathbb{E}\left[\langle\nabla L(W_t), W_{t+1} - W_t\rangle\right]$ and $\mathbb{E}\left[\|W_{t+1} - W_t\|^2\right]$ to Equation 6, we derive the following:

$$\mathbb{E}\left[\mathcal{L}(W_{t+1}) - \mathcal{L}(W_t)\right] \leq \mathbb{E}\left[\langle\nabla\mathcal{L}(W_t), W_{t+1} - W_t\rangle\right] + \frac{\beta}{2}\mathbb{E}\left[\|W_{t+1} - W_t\|^2\right]$$

$$\leq \underbrace{-\frac{\eta_t}{2}\mathbb{E}\left[\|\nabla\mathcal{L}(W_t)\|^2\right] - \frac{\eta_t}{2N}\sum_{i=1}^N\mathbb{E}\left[\left\|\nabla\mathcal{L}_i(W_t)\right\|^2\right]}_{A_1}$$

$$+ \underbrace{\frac{\eta_t}{4}\mathbb{E}\left[\|\nabla\mathcal{L}(W_t)\|^2\right] + \frac{2\epsilon\eta_t}{N}\sum_{i=1}^N\mathbb{E}\left[\|\nabla\mathcal{L}_i(W_t)\|^2\right] + 2\epsilon\eta_t^2\sigma_l^2}_{A_2}$$

$$+ \frac{2\beta(1+\epsilon)\eta_t^2}{N}\sum_{i=1}^N\mathbb{E}\left[\|\nabla\mathcal{L}_i(W_t)\|^2\right] + 2\beta(1+\epsilon)\eta_t^2\sigma_l^2$$

$$= -\frac{\eta_t}{4}\mathbb{E}\left[\|\nabla\mathcal{L}(W_t)\|^2\right] + \frac{\eta_t}{N}\underbrace{\left\{-\frac{1}{2} + 2\epsilon + 2\beta(1+\epsilon)\eta_t\right\}}_{\leq 0 \text{ if we choose } \eta_t \leq \frac{1-4\epsilon}{4\beta(1+\epsilon)}}\sum_{i=1}^N\mathbb{E}\left[\left\|\nabla\mathcal{L}_i(W_t)\right\|^2\right] + 2\eta_t^2(\epsilon + \beta + \beta\epsilon)\sigma_l^2$$

$$\leq -\frac{\eta_t}{4}\mathbb{E}\left[\|\nabla\mathcal{L}(W_t)\|^2\right] + 2\eta_t^2(\epsilon + \beta + \beta\epsilon)\sigma_l^2$$

Ultimately, by applying the telescoping sum over $t = 0, 1, \ldots, T - 1$, we arrive at the following result:

$$\mathcal{L}^* - \mathbb{E}\left[\mathcal{L}(W_0)\right] \leq \sum_{t=0}^{T-1} -\frac{\eta_t}{4} \mathbb{E}\left[\|\nabla\mathcal{L}(W_t)\|^2\right] + \sum_{t=0}^{T-1} 2\eta_t^2(\epsilon + \beta + \beta\epsilon)\sigma_l^2$$

In this case, $\mathcal{L}^*$ stands for the minimum of $\mathcal{L}(W)$.

By performing a division by $H_T = \sum_{t=0}^{T-1} \eta_t$ on both sides and utilizing some algebraic adjustments, we arrive at the following expression:

$$\frac{1}{4H_T}\sum_{t=0}^{T-1} \eta_t \mathbb{E}\left[\|\nabla\mathcal{L}(W_t)\|^2\right] \leq \frac{\mathbb{E}\left[\mathcal{L}(W_0)\right] - \mathcal{L}^*}{H_T} + 2(\epsilon + \beta + \beta\epsilon)\sigma_l^2\left(\frac{1}{H_T}\sum_{t=0}^{T-1} \eta_t^2\right) \quad (7)$$

With a decreasing learning rate such as $\eta_t = \frac{\eta_0}{t+1}$, we observe that $H_T = \sum_{t=0}^{T-1} \eta_t$ tends towards infinity as $T$ grows, while $\sum_{t=0}^{T-1} \eta_t^2$ remains bounded. Therefore, as $T \to \infty$, the upper bound in Equation 7 converges to 0, confirming the convergence to a stationary point. $\qquad\square$

# E COMPLEXITY ANALYSIS AND TRADE-OFFS

Although the MAPA approach is advantageous for communication efficiency by having a large matrix $A$, it may pose challenges for devices with limited memory and computational resources. $\text{MAPAX}_k$ provides a trade-off that can reduce memory consumption at the expense of some communication efficiency by partitioning the update vector $W$ and factorizing each part separately, making it a customizable solution for resource-constrained devices. In the following, we show the computation of memory, time, computation, communication, expected error rate, and error variance of $\text{MAPAX}_k$. Finally, we summarize the results in Table 2 and how, in practice, the $\text{MAPAX}_k$ can be tuned to address the client constraint.

**Memory Complexity**: The additional memory complexity opposed by $\text{MAPAX}_k$ comes mainly from storing a large reconstruction matrix $A$, as the model gradient is compressed in matrix $B$, which is a reduction in memory compared to traditional FL.

Let $\Delta W \in \mathbb{R}^{\frac{d}{k} \times k}$ be the update matrix of a model, which $\text{MAPAX}_k$ factorizes $\Delta W = AB$, where $A \in \mathbb{R}^{\frac{d}{k} \times p}$ and $B \in \mathbb{R}^{p \times k}$. Therefore, the additional memory overhead can be computed as:

$$Memory = O(\frac{dp}{k}).$$

**Communication Overhead** The communication overhead solely depends on the size of matrix $B$, therefore regardless of batching for one FL round, the communication cost will be as:

$$Comm = O(pk)$$

**Error Rate and Variance** As the results of Definition 2 and Proposition 3, the error rate and variance can be defined as:

$$\mathbb{E}[E] = 1 - \frac{pk}{d} \quad ; \quad \mathbf{Var}[E] = \frac{2p(\frac{d}{k} - p)}{(\frac{d}{k})^2(\frac{d}{k} + 2)} = \frac{2k^2 p(d - pk)}{d^2(d + 2k)}.$$

**Tuning Parameters** In practice, given a model with constant $d$ parameters, we explore the strategy of setting the tunable parameters $p$ and $k$ to meet the client's resource constraints.

First, clients should decide on a trade-off between the communication bandwidth and tolerance for error, as both factors are related to the $pk$ term. Therefore, setting $pk = c$ for a constant $c$ is recommended. Given constant $pk = c$ and $k \ll d$, we can rewrite memory complexity and approximate variance as:

$$Memory = O(\frac{dc}{sk^2})$$

$$\mathbf{Var}[E] = \frac{2kc(d - c)}{d^2(d + 2k)} \approx \frac{2c(d - c)}{d^3} k.$$

Therefore, clients should decide on a trade-off between memory and tolerance of error variance, as both factors relate to the $k$. It is important that all clients agree on the values for $p$ and $k$ to ensure the consistency of the updates during FL rounds.

Table 2: Complexity Analysis and Trade-offs for $\text{MAPAX}_k$

| Aspect | Expression | Description |
|---|---|---|
| Memory Complexity | $O\left(\frac{dp}{k}\right)$ | Additional memory for storing matrix $A$. |
| Communication Overhead | $O(pk)$ | Communication cost per FL round. |
| Expected Error Rate | $\mathbb{E}[E] = 1 - \frac{pk}{d}$ | Error rate depends on $pk$ and $d$. |
| Error Variance | $\mathbf{Var}[E] = \frac{2k^2 p(d-pk)}{d^2(d+2k)}$ | Variance as a function of $p$, $k$, and $d$. |
| Tunable Parameters | $pk = c$ | set $c$ based on bandwidth and error trade-offs. |
| Memory with $pk = c$ | $O\left(\frac{dc}{k^2}\right)$ | Memory as a function of $k$. |
| Error Variance with $pk = c$ | $\mathbf{Var}[E] \approx \frac{2c(d-c)}{d^3} k$ | Variance as a function of $k$. |

## F  IID AND CLIENT SAMPLING

This section includes the results of additional experiments on IID distribution and client sampling for MNIST, FMNIST, CIFAR-10, and CIFAR-100.

| **MNIST Maximum Accuracy and Communication Cost** | | | | | | | |
|---|---|---|---|---|---|---|---|
| | IID | | | | NON-IID | | |
| | All clients | | 10% of clients | | All clients | | 10% of clients | |
| **Method** | Com. | Acc | Com. | Acc | Com. | Acc | Com. | Acc |
| FedAvg | 100% | 99.6% | 100% | 99.5% | 100 % | 98.9% | 100% | 97.6% |
| Sparse | 10.1% | 97.7% | 12.3% | 97.5% | 15.3% | 93.8% | 17.3% | 90.2% |
| Quantize | 9.3% | 98.8% | 10.5% | 98.7% | 31.3% | 97.6% | 33.1% | 96.1% |
| EvoFed | 8.5% | 98.6% | 8.8 % | 98.3% | 9.4 % | **98.5%** | 10.3% | 97.1% |
| FA-LoRA | 20.3% | 97.4% | 22.2% | 97.2% | 30.2% | 97.0% | 37.3% | 95.3% |
| **MAPAX**$_{d/64}$ | 2.1% | 98.8% | 2.5% | 98.6% | 3.1% | 98.1% | 3.5% | 97.5% |
| **MAPAX**$_{d/256}$ | 2.0% | 98.8% | 2.2% | 98.7% | 3.0% | 98.2% | 3.2% | 97.7% |
| **MAPAX**$_{d/1024}$ | **1.6%** | 98.9% | **1.9%** | 98.7% | **2.9%** | 98.5% | **3.0%** | **97.8%** |
| **MAPA (our)** | **1.6%** | **98.9%** | **1.9%** | **98.8**% | **2.9%** | 98.5% | **3.0%** | **97.8%** |

Table 3: All baselines performance on MNIST dataset with IID and non-IID distribution for both client sampling of 100% and 10%.

| **FMNIST Maximum Accuracy and Communication Cost** | | | | | | | |
|---|---|---|---|---|---|---|---|
| | IID | | | | NON-IID | | |
| | All clients | | 10% of clients | | All clients | | 10% of clients | |
| **Method** | Com. | Acc | Com. | Acc | Com. | Acc | Com. | Acc |
| FedAvg | 100% | 92.7% | 100% | 92.2% | 100% | 89.2% | 100% | 87.3% |
| Sparse | 16.0% | 84.4% | 18.4% | 83.9% | 24.1% | 81.1% | 26.3% | 78.6% |
| Quantize | 14.7% | 83.6% | 16.1% | 83.2% | 24.1% | 78.7% | 25.8% | 79.3% |
| EvoFed | 6.8% | 90.4% | 7.3% | 90.0% | 7.6% | 87.7% | 8.5% | 85.9% |
| FA-LoRA | 11.5% | 87.9% | 12.7% | 87.5% | 17.9% | 84.1% | 20.1% | 81.5% |
| **MAPAX**$_{d/64}$ | 2.3% | 91.0% | 2.7% | 90.7% | 3.4% | 87.7% | 3.8% | 85.9% |
| **MAPAX**$_{d/256}$ | 2.1% | 91.3% | 2.5% | 91.0% | 3.3% | 87.9% | 3.7% | 86.1% |
| **MAPAX**$_{d/1024}$ | 1.9 % | 91.2 % | **2.2%** | 91.1 % | **3.1%** | 87.9 % | 3.5 % | 86.3 % |
| **MAPA (our)** | **1.8%** | **91.4%** | **2.2%** | **91.3**% | **3.1%** | **88.0%** | **3.4%** | **86.5**% |

Table 4: All baselines performance on FMNIST dataset with IID and non-IID distribution for both client sampling of 100% and 10%.

| **CIFAR-10 Maximum Accuracy and Communication Cost** | | | | | | | |
|---|---|---|---|---|---|---|---|
| | IID | | | | NON-IID | | |
| | All clients | | 10% of clients | | All clients | | 10% of clients | |
| **Method** | Com. | Acc | Com. | Acc | Com. | Acc | Com. | Acc |
| FedAvg | 100% | 89.8% | 100% | 88.5% | 100% | 65.1% | 100% | 62.8% |
| Sparse | 1.1% | 63.1% | 1.3% | 62.6% | 1.0% | 47.1% | 1.2% | 46.5% |
| Quantize | 6.2% | 84.8% | 6.7% | 84.3% | 5.0% | 67.1% | 5.4% | 66.3% |
| EvoFed | 2.0% | 65.9% | 2.3% | 65.3% | 1.9% | 48.9% | 2.2% | 48.1% |
| FA-LoRA | 1.3% | 69.0% | 1.5% | 68.5% | 1.1% | 49.2% | 1.4% | 48.5% |
| **MAPAX**$_{d/64}$ | 1.2% | 88.7% | 1.5% | 88.2% | 1.1% | **68.2%** | 1.3% | 67.6% |
| **MAPAX**$_{d/256}$ | 1.1% | 88.8% | 1.4% | 88.3% | **1.0%** | **68.2%** | 1.2% | 67.8% |
| **MAPAX**$_{d/1024}$ | **1.0%** | 88.8% | **1.3%** | 88.4% | **1.0%** | **68.2%** | **1.1%** | **68.0%** |
| **MAPA (our)** | **1.0%** | **88.9%** | **1.3%** | **88.5%** | **1.0%** | **68.2%** | **1.1%** | **68.1%** |

Table 5: All baselines performance on CIFAR-10 dataset with IID and non-IID distribution for both client sampling of 100% and 10%.

| CIFAR-100 Maximum Accuracy and Communication Cost | | | | | | | |
|---|---|---|---|---|---|---|---|
| | IID | | | | NON-IID | | |
| | All clients | | 10% of clients | | All clients | | 10% of clients | |
| Method | Com. | Acc | Com. | Acc | Com. | Acc | Com. | Acc |
| FedAvg | 100% | 42.1% | 100% | 41.6% | 100% | 18.0% | 100% | 16.2% |
| Sparse | 7.0% | 35.8% | 8.5% | 34.5% | 7.5% | 12.1% | 8.1% | 10.8% |
| Quantize | 54.0% | 32.1% | 56.1% | 31.6% | 54.2% | 10.2% | 55.8% | 9.6% |
| EvoFed | 0.9% | 36.3% | 1.1% | 35.9% | 0.2% | 16.5% | 0.3% | 15.6% |
| FA-LoRA | 1.2% | 34.7% | 1.4% | 33.9% | 0.2% | 14.1% | 0.3% | 13.5% |
| **MAPAX**$_{d/64}$ | 0.3% | 36.6% | 0.4% | 36.1% | 0.1% | 16.8% | 0.2% | 16.2% |
| **MAPAX**$_{d/256}$ | 0.2% | 36.6% | 0.3% | 36.2% | 0.09% | 16.8% | 0.1% | 16.3% |
| **MAPAX**$_{d/1024}$ | **0.08%** | **36.7%** | **0.1%** | **36.5%** | **0.08%** | **16.8%** | **0.09%** | **16.4%** |
| **MAPA (our)** | **0.08%** | **36.7%** | **0.1%** | **36.5%** | **0.08%** | **16.8%** | **0.09%** | **16.4%** |

Table 6: All baselines performance on CIFAR-100 dataset with IID and non-IID distribution for both client sampling of 100% and 10%.

## G  MODEL ARCHITECTURES AND HYPERPARAMETERS

### NEURAL NETWORK ARCHITECTURE

The model configuration and training used in this work are provided in Table 7 and 8.

| Parameter | MNIST | FMNIST | CIFAR-10 | CIFAR-100 |
|---|---|---|---|---|
| Network Name | CNN | CNN | CNN | ResNet |
| Number of Convolutional Layers | 2 | 2 | 3 | 2 |
| Features in 1st Block | 8 | 8 | 64 | 64 |
| Features in 2nd Block | 16 | 16 | 128 | 64 |
| Kernel Size (Layer 1) | 5x5 | 5x5 | 5x5 | 5x5 |
| Kernel Size (Layer 2) | 5x5 | 5x5 | 5x5 | 5x5 |
| Stride (Layer 1) | 1 | 1 | 1 | 1 |
| Stride (Layer 2) | 1 | 1 | 1 | 1 |
| Number of Linear Layers | 1 | 1 | 2 | 2 |
| Features in Hidden Layers | 1 | 1 | 256 | 128 |
| Number of Output Units | 10 | 10 | 10 | 10 |

Table 7: Neural Network Configuration

### TRAINING HYPERPARAMETERS

The training was performed with the following key hyperparameters:

| Parameter | MNIST | FMNIST | CIFAR-10 | CIFAR-100 |
|---|---|---|---|---|
| Batch Size | 32 | 32 | 32 | 32 |
| Optimizer | SGD | SGD | SGD | SGD |
| Learning Rate | 0.00594 | 0.00594 | 0.0041 | 0.0041 |
| L1 Regularization | 0.0003 | 0.0003 | 0.0001 | 0.0001 |
| L2 Regularization | 0.004 | 0.004 | 0.002 | 0.002 |

Table 8: Training Hyperparameters

