# OpenReview forum: "Communication-Efficient Federated Learning via Model-Agnostic Projection Adaptation"
_ICLR.cc/2025/Conference — Submitted to ICLR 2025_

### Official Review · Reviewer_F353 · 2024-11-02

**Soundness:** 3
**Presentation:** 2
**Contribution:** 2
**Rating:** 5
**Confidence:** 4

**Summary:**

This paper introduces a method that factorizes the model within the parameter space, aiming to reduce communication overhead in the FL setting. By transmitting only the lower-dimensional projection vector rather than the entire parameter vector, the approach reduces the parameter need to be transmitted

**Strengths:**

1. Compared to related approaches like FFA-LoRA, this work’s MAPA method applies LoRA directly in the parameter space, thereby improving flexibility for agnostic models.

2. The extended method, MAPAX, provides a valuable trade-off between computational constraints and communication overhead. This balance makes MAPAX an interesting direction for further exploration.

**Weaknesses:**

**Algorithm design**

1. The method requires generating and storing a matrix $A$ with $k \times d$ entries, where $d$ represents the number of parameters in the entire model, which is typically very large. This raises significant concerns about the method's practicality due to the immense computational and memory requirements. Although the authors introduce MAPAX as a solution to mitigate these issues, there is a lack of experimental results validating its effectiveness. Without empirical evidence, it's challenging to assess whether MAPAX successfully addresses the computational and storage challenges posed by MAPA.

2. The matrix $A$ is generated randomly, similar to the approach in FAA-LoRA [1]. Recent studies [2] have indicated that random generation of such matrices may lead to suboptimal performance.

**Theoretical analysis**

1. The "representation capacity rate" introduction in Definition 2 lacks clarity and justification. The term "representation capacity" is not well-defined in the context of the paper, and it's unclear why the proposed quantity effectively measures it. There is no evident connection between the introduced definition and established metrics for the complexity of function classes, such as covering numbers.

2.  In Definition 2, the focus is solely on the rank $k$ of matrix $A$, neglecting the influence of the dimension $d$. In representation learning literature, the complexity typically depends on both $d$ and $k$.

3.  Due to the unclear definition of the representation capacity rate, the claim in Proposition 1 that the proposed method "...achieves a higher representation capacity..." lacks substantiation.

4.  The convergence analysis is built upon Assumption 4, which appears to be too restrictive and not reflective of practical scenarios. For instance, this assumption may not hold if $W_t$ reaches a saddle point in the non-convex optimization landscapes.

**Experiments**

1. The reported accuracy on the CIFAR-10 dataset is unexpectedly low. Previous work, such as FedRep [3], published in 2021, has demonstrated significantly higher accuracy levels, even though it is not SOTA.  Besides, the selection of baseline algorithms in the experimental evaluation is inadequate.

2. Although MAPAX is proposed to address the computational and memory burdens introduced by MAPA, the paper does not present any experimental results for it.

References:

[1] Sun, Youbang, et al. "Improving loRA in privacy-preserving federated learning." arXiv preprint arXiv:2403.12313 (2024).

[2] Guo, Pengxin, et al. "Selective Aggregation for Low-Rank Adaptation in Federated Learning." arXiv preprint arXiv:2410.01463 (2024).

[3] Collins, Liam, et al. "Exploiting shared representations for personalized federated learning." International conference on machine learning. PMLR, 2021.

**Questions:**

See the "Weakness" section.

---

> ### Author Response · Authors · 2024-11-23
>
> **1. Computational and memory concerns:**
> > The method requires generating and storing a matrix ... with entries, where ... represents the number of parameters in the entire model, which is typically very large.
>
> We appreciate the reviewer's observation and acknowledge the concerns regarding the large reconstruction matrix. First, we want to note that MAPA's memory complexity (although very high) aligns with many prior works in this area and is not unusual.
> For instance, [4,5] store random matrices, and [6,7,8] store previous gradient signals that have a similar shape to MAPA's $A$.
>
> Based on your concern, we updated the manuscript to emphasize that MAPAX demonstrates performance comparable to MAPA, as supported by additional experimental results and theoretical justification provided in the revised manuscript. Therefore, our partitioning approach (MAPAX) significantly reduces memory usage while maintaining similar performance, making MAPA accessible to low-resource clients.
>
> Please refer to the updated Table 1, Figures 6, and Appendix F for empirical results comparing MAPAX and Proposition 3 and Appendix E for the theoretical analysis.
>
> ---
>
> **2. Random matrix generation:**
> > The matrix is generated randomly, similar to the approach in FAA-LoRA [1]. Recent studies [2] have indicated that random generation of such matrices may lead to suboptimal performance.
>
> Thanks for pointing this out. We also acknowledged that the FFA-LoRA frozen random matrix is suboptimal and provided evidence in Appendix B.
> Regarding your comment, we have updated the text to emphasize that our matrix generation differs significantly from FFA-LoRA. Specifically, we generate a new matrix at each iteration, whereas FFA-LoRA uses a fixed and frozen matrix $A$ throughout the entire training process.
>
> This limitation in FFA-LoRA arises from the architectural differences compared to MAPA. In MAPA, $W$ is not inherently fixed during training; after obtaining the optimal $B$, we update $W_t = W_{t-1} + AB$, which allows us to generate a new matrix $A$ and reset $B$ to zero at each iteration. In contrast, FFA-LoRA techniques keep $W$ fixed, necessitating that matrix $A$ remain unchanged to preserve training stability and use matrix $B$ to accumulate training adaptation. Consequently, our approach allows for continuous change in $A$, which prevents suboptimal performance, as also demonstrated in Appendix B of the initial submission.
>
> Regarding the [2] solution, while this paper provides valuable insights into the problem, it is not directly applicable to communication-efficient training in federated learning since they train and communicate the matrix $A$, undermining the ability to achieve communication efficiency. We have updated the manuscript (Lines 171–181) to acknowledge the findings of [2] while presenting our proposed solution and details in Appendix B.
>
> ---
>
> **3. Representation capacity rate:**
> > The ‘representation capacity rate’ introduction in Definition 2 lacks clarity and justification.
>
> We agree with you on this point and have clarified our intentions for this definition. We provided a new definition of **representation certainty**, which revolves around error rate and variance, and updated the propositions accordingly. Importantly, this update does not impact our methodology, results, convergence analysis, motivation, or the intuition behind our solution; it only revises the theoretical explanation of MAPA and MAPAX.
> We would be happy to know if you find the current version of the theoretical analysis satisfactory.
>
> ---
>
> **4. Convergence analysis and Assumption 4:**
> >The convergence analysis is built upon Assumption 4, which appears too restrictive and not reflective of practical scenarios.
>
> We appreciate your constructive feedback. We relaxed Assumption 4 to have an unbiased estimator instead of a bounded error. We addressed concerns using the Johnson–Lindenstrauss lemma to justify Assumption 4. Additional details are provided in Appendix C.
>
> **References:**
> [4] Shi, Zai, and Atilla Eryilmaz. "Communication-efficient Subspace Methods for High-dimensional Federated Learning." 2021 17th International Conference on Mobility, Sensing and Networking (MSN). IEEE, 2021.
> [5] Mahdi Rahimi, Mohammad, et al. "EvoFed: Leveraging Evolutionary Strategies for Communication-Efficient Federated Learning." arXiv e-prints (2023): arXiv-2311.
> [6] Azam, Sheikh Shams, et al. "Recycling model updates in federated learning: Are gradient subspaces low-rank?." International Conference on Learning Representations. 2021.
> [7] Oh, Yongjeong, et al. "Vector quantized compressed sensing for communication-efficient federated learning." 2022 IEEE Globecom Workshops (GC Wkshps). IEEE, 2022.
> [8] Park, Sangjun, and Wan Choi. "Regulated subspace projection based local model update compression for communication-efficient federated learning." IEEE Journal on Selected Areas in Communications 41.4 (2023): 964-976.

---

> > ### Author Response · Authors · 2024-11-23
> >
> > **5. CIFAR-10 experimental results:**
> > >The reported accuracy on the CIFAR-10 dataset is unexpectedly low. Previous work, such as FedRep [3], published in 2021, has demonstrated significantly higher accuracy levels...
> >
> > Thank you for the notes on the CIFAR-10 performance and for highlighting the FedRep paper. Upon further investigation, we identified three reasons for the lower performance, which we address here:
> >
> > **Personalization vs. Adaptation:**
> > FedRep primarily focuses on personalization, and the reported accuracy reflects client-specific performance on their respective test data, which shares a similar distribution with their training data (i.e., the same classes), while we report the global performance across all classes.
> >
> > We have updated Table 1 to include local (personalization) accuracy, which allows for a direct comparison to FedRep. Our results show that our method and baselines perform similarly to FedRep's baseline + FT (fine-tuning) regarding local accuracy. However, unlike FedRep, which focuses solely on local performance, our primary objective is to improve global accuracy, ensuring effective adaptation of the global model to all clients.
> >
> > **Compression and Communication Efficiency:**
> > Our primary objective is to achieve comparable global model performance to FedAvg while significantly reducing communication overhead, not suppressing FedAvg performance. This is not the main focus of FedRep, which trains and fine-tunes private client-specific heads to improve local performance. Since communication efficiency is central to our work, we focus less on absolute accuracy values and more on the trade-offs between performance degradation and communication benefits relative to FedAvg.
> >
> > **Preprocessing:**
> > FedRep applies additional preprocessing steps to the training data that can enhance training performance. We repeated our experiments regarding CIFAR-10, incorporating these preprocessing steps to get similar baselines.
> >
> > Given these factors, we have included additional data in Table 1 to clarify the primary goals of our evaluation, and we have reflected on this data throughout the manuscript.
> >
> > **Updated experimental results in comparison to [3]:**
> >
> > | **Method**               | **Com. Cost** | **Local Acc. (cifar-10)** | **Global Acc. (cifar-10)** | **Com. Cost** | **Local Acc. (cifar-100)** | **Global Acc. (cifar-100)** |
> > |--------------------------|----------------|---------------------------|----------------------------|----------------|----------------------------|-----------------------------|
> > | FedAvg                                                                | 100%                   | 98.8%                        | 69.0%                          | 100%                   | 42.1%                          | 18.0%                          |
> > | FedAvg+FT [3]                                                         | ---                    | 87.65%                       | ---                             | ---                    | 55.44%                         | ---                             |
> > | FedRep [3]                                                            | ---                    | 87.70%                       | ---                             | ---                    | 56.70%                         | ---                             |
> > | MAPAX$_{d/64}$                                                  | 1.1%                   | 88.7%                        | 68.2%                          | 0.1%                   | 36.6%                          | 16.8%                          |
> > | MAPAX$_{d/256}$                                                 | 1.0%                   | 88.8%                        | 68.2%                          | 0.09%                  | 36.6%                          | 16.8%                          |
> > | MAPAX$_{d/1024}$                                                | 1.0%                   | 88.8%                        | 68.2%                          | 0.08%                  | 36.7%                          | 16.8%                          |
> > | MAPA (our)                                                            | 1.0%                   | 88.9%                        | 68.3%                          | 0.08%                  | 36.7%                          | 16.8%                          |
> >
> > **Table a.** CIFAR-10 and CIFAR-100 performance summary. Communication cost and global accuracy for FedAvg+FT and FedRep are not applicable, while local accuracy values are provided.

---

> ### Author Response · Authors · 2024-11-23
>
> **6. Baselines:**
> >Besides, the selection of baseline algorithms in the experimental evaluation is inadequate.
>
> We added quantization as a new baseline for communication efficiency for all experiments. We want to emphasize that this paper's main objective and scope is to provide a communication-efficient algorithm for FL, so we evaluated baselines in the scope of communication efficiency and not regarding other aspects of FL.
>
> As most communication-efficient techniques can be combined and SOTA approaches offer a hybrid solution of sparsification and quantization, we believe it can be satisfactory to compare these fundamental ideas. In practice, one can effortlessly combine MAPA with quantization and sparsification or a hybrid of different techniques to achieve better performance.
>
> Regarding communication-efficient baselines, we currently provide sparsification and quantization as the most common fundamental approaches, alongside EvoFed and FFA-LoRA SOTA, which have similar methodologies to our work to highlight our method's advantages.
>
> ---
>
> **7. MAPAX performance:**
> >Although MAPAX is proposed to address the computational and memory burdens introduced by MAPA, the paper does not present any experimental results for it.
>
> We understand the importance of investigating MAPAX performance and representing the results more clearly. Based on all reviewers' comments, we dedicated Figure 6 to explicitly presenting MAPAX performance and analysis.
>
> Overall, Figure 6, Table 1, and Appendix F demonstrate that MAPAX can perform similarly to MAPA while using significantly less memory. Compared to MAPA, MAPAX's performance difference is less than 0.3%, and the additional communication burden does not exceed 1.0%.

---

> > ### Comment · Reviewer_F353 · 2024-11-26
> > **Thank you for the reply**
> >
> > Thank you for your rebuttal and the additional experiments. After carefully considering all the reviewers' concerns, I have decided to maintain my score.

---

> ### Author Response · Authors · 2024-11-26
>
> Thank you very much for your prompt response. We would greatly appreciate it if you could specify which concerns remain particularly relevant and which ones have been resolved. This will help us focus on addressing the remaining points effectively.
>
> Your timely feedback is invaluable and provides us with an opportunity to reflect on and refine our work. Thank you again for your support!

---

### Official Review · Reviewer_Z1ir · 2024-11-03

**Soundness:** 3
**Presentation:** 3
**Contribution:** 2
**Rating:** 5
**Confidence:** 3

**Summary:**

To reduce the substantial communication overhead in FL, this work presents MAPA and its extension MAPAX, a novel matrix factorization operating independently of the model architecture. The proposed method sends only projection vectors that leverage a shared reconstruction matrix generated at each round. MAPA treats the entire model parameter space as a single vector and factories it. The authors performed numerical experiments that showed that MAPA improves communication efficiency while maintaining strong model performance.

**Strengths:**

1. The paper provides theoretical analysis of MAPA's convergence and communication efficiency by clear and precise mathematical notations. The experiments on three dataset demonstrates MAPA's superiority over other methods.
2. The paper is well-organized, with straightforward writing and clear illustrations, especially the overview diagram, which effectively conveys MAPA's idea. Results are intuitively displayed, showcasing MAPA's improvement in reducing communication costs.

**Weaknesses:**

1. The paper lacks comprehensive experimental results for MAPAX, specifically missing accuracy and minimum cost outcomes on MNIST and CIFAR-10, which limits the ability to fully evaluate its effectiveness.
2. I am curious whether the proposed method can be applied to the IID setting or the scenario where clients are sampled in each communication round. Evaluating and clarifying MAPA's performance under these conditions would strengthen the paper.
3. Given the author's claim that MAPA is model-agnostic, the paper could strengthen this claim by combining MAPA with more FL models to demonstrate its versatility beyond the current implementations.
4. There are several presentation issues that can be addressed: a) the text is too small to read in Fig. 4; b). misalignment between L. 517-524 and Fig. 6. Also, it would be better to provide more analyses and explanations for Fig. 6 in the main paper.

**Questions:**

Refer to the weakness section.

---

> ### Author Response · Authors · 2024-11-23
>
> **1. Lack of diverse model architectures:**
> > Given the author's claim that MAPA is model-agnostic, the paper could strengthen this claim by combining MAPA with more FL models to demonstrate its versatility beyond the current implementations.
>
> We extended our experiments on CIFAR-100 using a ResNet architecture with 5.5 million parameters. Our study now includes three distinct architectures, ranging from 11,000 to 1.4 million and 5.5 million parameters. The experimental results demonstrate that MAPA scales effectively to larger tasks and diverse architectures and performs close to FedAvg, regardless of significantly less communication cost.
>
> The CIFAR-100 experimental results show that MAPA and MAPAX can achieve around 93.3% of the FedAvg performance while using nearly 1.0% of the communication. The full results can be found in Table 1 and Table 6 of Appendix F.
>
> ---
>
> **2. Missing MAPAX results:**
> > The paper lacks comprehensive experimental results for MAPAX, specifically missing accuracy and minimum cost outcomes on MNIST and CIFAR-10.
>
> Initially, Figure 6 was intended to illustrate various factorizations, including no factorization (FedAvg), LoRA, MAPA, and MAPAX. However, based on your comments and other reviewers' requests for a more in-depth analysis of MAPAX, we revised Figure 6 to focus exclusively on MAPAX results for FMNIST, providing a more detailed evaluation. Results for MNIST and CIFAR-10 datasets are now added to Figure 6.
>
> Figure 6, Table 1, and Appendix F demonstrate that MAPAX can perform similarly to MAPA while using significantly less memory. Compared to MAPA, MAPAX's performance difference is less than 0.3\%, and the additional communication burden does not exceed 1.0\%.
>
> ---
>
> **3. IID and sampled-client scenarios:**
> > I am curious whether the proposed method can be applied to the IID setting or the scenario where clients are sampled in each communication round.
>
> We conducted additional experiments on IID settings and client sampling, where 10% of clients are sampled per communication round. The results are in Appendix F, confirming MAPA's performance under these conditions. We appreciate your feedback, as these experiments can provide valuable insights into why MAPA outperforms other baselines.
>
> The IID and client sampling results can be summarized as follows:
>
> | **Dataset** | **Method** | **Communication Cost** | **Accuracy (MAPA / FedAvg)** |
> |-------------|------------|-------------------------|--------------------------------|
> | MNIST       | MAPA (our) | **1.6\%**              | **98.9\%** / 99.6\%          |
> | FMNIST      | MAPA (our) | **1.8\%**              | **91.4\%** / 92.7\%          |
> | CIFAR-10    | MAPA (our) | **1.0\%**              | **88.9\%** / 89.8\%          |
> | CIFAR-100   | MAPA (our) | **0.08\%**             | **36.7\%** / 42.1\%          |
>
> **Table a.** Performance summary in the IID setting for all clients.
>
> | **Dataset** | **Method** | **Communication Cost** | **Accuracy (MAPA / FedAvg)** |
> |-------------|------------|-------------------------|--------------------------------|
> | MNIST       | MAPA (our) | **1.9\%**              | **98.8\%** / 99.5\%          |
> | FMNIST      | MAPA (our) | **2.2\%**              | **91.3\%** / 92.2\%          |
> | CIFAR-10    | MAPA (our) | **1.3\%**              | **88.5\%** / 88.5\%          |
> | CIFAR-100   | MAPA (our) | **0.1\%**              | **36.5\%** / 41.6\%          |
>
> **Table b.** Performance summary in the IID setting for 10% of clients.
>
> These findings demonstrate that MAPA's strong performance in non-IID cases does not come only from capturing more comprehensive gradient information but also from better adaptation. We conclude this from the fact that various baselines perform much closer to MAPA in IID settings but fail to adapt as well as MAPA in non-IID settings.
>
> We can look at non-IID FL settings as continuously adaptive learning, which suggests that methods from adaptive learning, such as low-rank approximation, can be more effective.
>
> ---
>
> **4. Presentation issues:**
> > The text is too small to read in Fig. 4; misalignment between L. 517-524 and Fig. 6.
>
> We appreciate your attention to these details; we have improved the readability of Figure 4 and corrected the misalignment issues. Additionally, as you suggested, Figure 6 has been revised to focus exclusively on the MAPAX evaluation, providing a more thorough and detailed analysis to address reviewer concerns.

---

> > ### Author Response · Authors · 2024-12-01
> >
> > Dear Reviewer Z1ir,
> >
> > We appreciate your thoughtful and detailed feedback on our submission. Your comments have been instrumental in guiding significant improvements to the manuscript. We have carefully considered your observations and revised our work accordingly, particularly addressing:
> >
> > (i) Additional experiments to demonstrate MAPA’s versatility with diverse architectures,
> >
> > (ii) comprehensive evaluations of MAPAX, including accuracy and minimum cost outcomes for MNIST and CIFAR-10,
> >
> > (iii) experiments evaluating MAPA's performance in IID and sampled-client scenarios and
> >
> > (iv) resolving presentation issues to enhance clarity and alignment between text and figures.
> >
> > We are happy to answer any remaining concerns that you might have during the remainder of the Author-Reviewer discussion period, which ends in 2 days. Once again, thank you for your time and efforts.
> >
> > Best regards,
> >
> > Authors of "Communication-Efficient Federated Learning via Model-Agnostic Projection Adaptation"

---

### Official Review · Reviewer_E4jN · 2024-11-09

**Soundness:** 3
**Presentation:** 3
**Contribution:** 3
**Rating:** 5
**Confidence:** 5

**Summary:**

This paper proposes Model-Agnostic Projection Adaptation (MAPA) to address communication overhead in federated learning by factorizing the model parameter space as a single vector, independent of layer count or model architecture. MAPA splits the update into a fixed reconstruction matrix and a trainable projection vector, sending only the projection vector to the server. This reduces communication costs while allowing a flexible, high-capacity reconstruction matrix. Experimental results show MAPA outperforms existing methods in both communication efficiency and model performance. Yet, aligning the experimental setup with model-agnostic principles remains an area for improvement.

**Strengths:**

1. The Model-Agnostic Projection Adaptation (MAPA) method introduces a novel approach to reduce communication overhead in federated learning (FL) by factorizing the entire model parameter space. The method employs a fixed reconstruction matrix and a shared random seed, allowing for larger reconstruction matrices and smaller projection vector dimensions. The paper provides strong theoretical support.
2. By transmitting only the trainable projection vectors, MAPA addresses a major challenge in FL by cutting communication costs.
3. Experimental results indicate that MAPA outperforms existing FL methods in both communication efficiency and model performance.

**Weaknesses:**

1. Although the method is described as 'model-agnostic,' it currently uses the same CNN model and identical initial weights across all clients in the first round. This setup could contradict the claim of model agnosticism by implying a reliance on a specific model structure and initialization. Therefore, it is essential to either revise the claims or provide further evidence that MAPA can operate with diverse initializations and model types across clients.
2. Additional experiments with different model architectures are recommended to showcase the method’s versatility and robustness.
3. Further testing on more complex datasets, such as ImageNet, would help demonstrate MAPA’s scalability.

**Questions:**

NA

---

> ### Author Response · Authors · 2024-11-23
>
> **1. Model-agnostic claim:**
> > Although the method is described as 'model-agnostic,' it currently uses the same CNN model and identical initial weights across all clients in the first round. This setup could contradict the claim of model agnosticism...
>
> We appreciate the reviewer's observation and would like to clarify that our intention for the term "model-agnostic" in our work refers explicitly to the flexibility of the factorization framework, which operates on the parameter space as a whole, independent of the network's specific architecture or layer structure. This independence allows MAPA to generalize to various model types without requiring specialized adaptations for each architecture.
>
> Our work does not address heterogeneity in model architectures or initializations across clients, as this is orthogonal to MAPA's contributions. Instead, the framework focuses on enabling efficient communication through black-box factorization, which remains valid regardless of the specific client setup.
>
> This is due to the fact that we primarily focus on minimizing the reconstruction error as $\|\Delta W - AB\|_2^2$, which, given $A$ drawn from a random Gaussian distribution, spans the sub-space uniformly and ignores existing structure or correlation in $\Delta W$. This makes it a general black-box factorization technique.
>
> ---
>
> **2. Additional experiments on model architectures:**
> > Additional experiments with different model architectures are recommended to showcase the method's versatility and robustness.
>
> We extended our experiments on CIFAR-100 using a ResNet architecture with 5.5 million parameters.
> Our study now includes three distinct architectures, ranging from 11,000 to 1.4 million and 5.5 million parameters. The experimental results demonstrate that MAPA scales effectively to larger tasks and diverse architectures and performs close to FedAvg, regardless of significantly less communication cost.
>
> The CIFAR-100 experimental results show that MAPA and MAPAX can achieve around 93.3% of the FedAvg performance while using only nearly 1.0% of the communication. The full results can be found in Table 1 and Table 6 of Appendix F.
>
> ---
>
> **3. Dataset scalability:**
> >Further testing on more complex datasets, such as ImageNet, would help demonstrate MAPA's scalability.
>
> To demonstrate scalability, we performed additional experiments on CIFAR-100. The results in Table 1 and Appendix F highlight MAPA's effectiveness in handling complex datasets, achieving 93.3% of FedAvg performance while using nearly 1.0% of communication. While conducting extensive experiments on ImageNet would be ideal, given the short review timeline, it is beyond our available resources. If the current results are deemed satisfactory, we are committed to adding additional dataset experiments for the camera-ready version.

---

> > ### Author Response · Authors · 2024-12-01
> >
> > Dear Reviewer E4jN,
> >
> > We appreciate your thoughtful and constructive feedback on our submission. Your insights have been invaluable in improving the clarity and depth of our manuscript. We have carefully considered your comments and revised our work accordingly, particularly addressing the following:
> >
> > (i) the clarification of the "model-agnostic" claim,
> >
> > (ii) additional experiments showcasing MAPA's performance for larger model and dataset
> >
> > (iii) Further analysis of MAPA’s scalability and communication efficiency.
> >
> > We are happy to answer any remaining concerns that you might have during the remainder of the Author-Reviewer discussion period, which ends in 2 days. Once again, thank you for your time and efforts.
> >
> > Best regards,
> >
> > Authors of "Communication-Efficient Federated Learning via Model-Agnostic Projection Adaptation"

---

### Author Response · Authors · 2024-11-23

We sincerely thank all reviewers for their constructive feedback and insightful comments, which have greatly enhanced the quality and clarity of our manuscript.

We are particularly grateful for the reviewers' recognition of the novelty (E4jN), strong theoretical support (E4jN, Z1ir), experimental results (E4jN, Z1ir), clarity of diagrams and results (Z1ir), flexibility of MAPA (F353), and innovative and valuable idea of MAPAX (F353), all of which aim to address a major challenge in federated learning (FL) (E4jN).

Below, we outline the key improvements made to the manuscript and address specific concerns raised by each reviewer.

**Key improvements in the revised manuscript:**
- Inclusion of experiments with additional baselines, models, datasets, IID distribution, and client sampling scenarios.
- Revised theoretical justifications for improving clarity and added complexity analysis.
- In-depth analysis of MAPAX results and factorization.
- Addressed minor presentation issues for improved readability.

---

### Meta-Review · Area_Chair_Fv2G · 2024-12-19

**Metareview:**

This paper introduces the MAPA approach (and its extension MAPAX), which leverages the idea of matrix factorization technique to address the issue of communication overhead in federated learning.
Numerical experiments on commonly used datasets (e.g., MNIST, FMNIST, CIFAR-10, and CIFAR-100) are provided to demonstrate the superior communication efficiency of the proposed method.

While the reviewers agree that the paper is clearly written and that the theoretical results are of interest, I personally find the theoretical contributions less compelling (and so does Reviewer F353). The analysis does not attempt to improve existing methodologies or provide novel insights into a family of methods; instead, it narrowly focuses on the proposed method itself. As a consequence, **I do not think that the major contribution of this paper is of theoretical nature**.

The reviewers also raise significant concerns regarding the lack of comprehensive empirical support. These include the need for experiments on different model architectures and larger datasets such as ImageNet (raised by Reviewer E4jN), evaluation with more federated learning (FL) approaches to validate the “model-agnostic” claim (raised by Reviewer Z1ir), and comparisons with additional baselines (raised by Reviewer F353).

This is a borderline paper. During the rebuttal phase, the authors actively engaged with the reviewers, resulting in improvements to the paper. However, despite these productive interactions, the reviewers remain unconvinced about the significance of the algorithmic and empirical contribution. As a result, I recommend rejecting the paper.

**Additional Comments On Reviewer Discussion:**

The reviewers raised the following concerns:

- Theoretical statements lack precision (raised by Reviewer F353): This issue, in my opinion, has been **successfully addressed** by the authors. However, I still find it difficult to argue that the theoretical contributions of this paper are significant for the reasons mentioned above.
- Lack of comprehensive numerical results: This includes the need for experiments on different model architectures and larger datasets such as ImageNet (raised by Reviewer E4jN), evaluation with additional federated learning (FL) approaches to validate the `model-agnostic` claim (raised by Reviewer Z1ir), and comparisons with more baseline methods (raised by Reviewer F353).
These concerns have only been **partially resolved** by the authors, leaving significant room for further improvement.

I have carefully considered all of the above points in making my final decision.

---

### Decision · Program_Chairs · 2025-01-22

Reject